

# Simulating secondary organic aerosol in a regional air quality model using the statistical oxidation model – Part 3: Assessing the influence of semi-volatile and intermediate volatility organic compounds and NO$_X$

Ali Akherati[1], Christopher D. Cappa[2], Michael J. Kleeman[2], Kenneth S. Docherty[3], Jose L. Jimenez[4], Stephen M. Griffith[5], Sebastien Dusanter[6], Philip S. Stevens[5], and Shantanu H. Jathar[1]

[1]Department of Mechanical Engineering, Colorado State University, Fort Collins, CO, USA
[2]Department of Civil and Environmental Engineering, University of California Davis, Davis, CA, USA
[3]Jacobs Technology, Raleigh, NC, USA
[4]Department of Chemistry and Cooperative Institute for Research in Environmental Sciences (CIRES), University of Colorado, Boulder, CO, USA
[5]School of Public and Environmental Affairs and Department of Chemistry, Indiana University, Bloomington, IN, USA
[6]IMT Lille Douai, Univ. Lille, SAGE - Département Sciences de l'Atmosphère et Génie de l'Environnement, 59000 Lille, France
*Correspondence to*: Shantanu H. Jathar (shantanu.jathar@colostate.edu)

## Abstract

Semi-volatile and intermediate-volatility organic compounds (SVOCs and IVOCs) from anthropogenic sources are likely to be important precursors of secondary organic aerosol (SOA) in urban airsheds yet their treatment in most models is based on limited and obsolete data, or completely missing. Additionally, gas-phase oxidation of organic precursors to form SOA is influenced by the presence of nitric oxide (NO), but this influence is poorly constrained in chemical transport models. In this work, we updated the organic aerosol model in the UCD/CIT chemical transport model to include (i) a semi-volatile and reactive treatment of primary organic aerosol (POA), (ii) emissions and SOA formation from IVOCs, (iii) the NO$_X$ influence on SOA formation, and (iv) SOA parameterizations for SVOCs and IVOCs that are corrected for vapor wall loss artifacts during chamber experiments. All updates were implemented in the statistical oxidation model (SOM) that simulates the oxidation chemistry, thermodynamics, and gas/particle partitioning of organic aerosol (OA). Model treatment of POA, SVOCs, and IVOCs was based on an interpretation of a comprehensive set of source measurements and resolved broadly by source type. The NO$_X$ influence on SOA formation was calculated offline based on measured and modeled VOC:NO$_X$ ratios. And finally, the SOA formation from all organic precursors (including SVOCs and IVOCs) was modeled based on recently derived parameterizations that accounted for vapor wall loss artifacts in chamber experiments. The updated model was used to simulate a two week summer episode over southern California at a model resolution of 8 km.

When combustion-related POA was treated as semi-volatile, modeled POA mass concentrations were reduced by 30-50% in the urban areas in southern California but were still too high when compared against measurements made at Riverside, CA during the Study of Organic Aerosols at Riverside (SOAR-1) campaign of 2005. Treating all POA (except that from marine sources) to be semi-volatile resulted in a larger reduction in POA mass concentrations and allowed for a better model-measurement comparison at Riverside. Model predictions suggested that both SVOCs (evaporated POA vapors) and IVOCs did not contribute significantly to SOA mass concentrations in the urban areas (<5% and <15% of the total SOA respectively) as the timescales for SOA production appeared to be shorter than the timescales for transport out of the urban airshed. Comparisons of modeled IVOC concentrations with measurements of anthropogenic SOA precursors in southern California seemed to imply that IVOC emissions were underpredicted in our updated model by a factor of 2. We suspect that these missing IVOCs might arise from the use of volatile chemical products such as pesticides, coatings, cleaning agents, and personal care products. Correcting for the vapor wall loss artifact in chamber experiments enhanced SOA mass concentrations although the enhancement was precursor- as well as NO$_X$-dependent. Accounting for the influence of NO$_X$ using the VOC:NO$_X$ ratios resulted in





better predictions of OA mass concentrations in rural/remote environments but still underpredicted OA mass concentrations in urban environments, potentially due to the missing urban emissions/chemical source of OA. Finally, simulations performed for the year 2035 showed that despite reductions in VOC and $NO_X$ emissions in the future, SOA mass concentrations may be higher than in the year 2005, primarily from increased hydroxyl radical (OH) concentrations due to lower ambient $NO_2$ concentrations.

**Glossary**

OA - Organic aerosol

POA - Primary organic aerosol or direct emissions of organic aerosol

SOA - Secondary OA or organic aerosol formed in the atmosphere

VOC - Volatile organic compound

NMOG - Non-methane organic gas

SVOC - Semi-volatile organic compound

IVOC - Intermediate-volatility organic compound

HOA - Hydrocarbon-like organic aerosol measured by the aerosol mass spectrometer

OOA - Oxygenated organic aerosol measured by the aerosol mass spectrometer

aV-SOA - Anthropogenic SOA formed from VOC oxidation

bV-SOA - Biogenic SOA formed from VOC oxidation

aS-SOA - Anthropogenic SOA formed from SVOC oxidation

aI-SOA - Anthropogenic SOA formed from IVOC oxidation

**1 Introduction**

Organic aerosol (OA) is an important yet uncertain component of atmospheric aerosol (Fuzzi et al., 2015; Jimenez et al., 2009) and has large impacts on air quality, climate, and human health (Pachauri et al., 2014). Combustion sources such as motor vehicles, biomass burning, and food cooking are significant contributors to atmospheric OA from urban to regional to global scales (Bond et al., 2004). Yet, in urban environments where combustion emissions are a dominant source, atmospheric models often underpredict total OA mass concentrations (e.g., Carlton et al. (2010)). Models based on older parameterizations also predict much lower contributions of secondary organic aerosol in urban areas (e.g., Volkamer et al. (2006); Jathar et al. (2017a)), and may overemphasize the role of mobile sources (e.g., Ensberg et al. (2014)), suggesting that combustion-related OA and other urban sources may not be well represented in models. There is a need to improve the treatment of combustion-related OA in atmospheric models since these improvements will allow for better predictions of air quality that are needed to mitigate climate and health impacts from anthropogenic combustion sources, and will facilitate improved understanding of additional potentially missing sources.

Research over the past decade has made major inroads in understanding the sources and properties of combustion-related OA (Gentner et al., 2017). Combustion sources directly emit organic particles (primary organic aerosol, POA) and also emit gaseous organic compounds that are oxidized in the atmosphere to form secondary organic aerosol (SOA). A majority of the combustion-related POA mass is now understood to be semi-volatile, that is material that





exists in a dynamic equilibrium between the vapor and particle phases (Grieshop et al., 2009a, 2009b; Huffman et al., 2009; Kuwayama et al., 2015; Lipsky and Robinson, 2006; May et al., 2013a, 2013b, 2013c; Robinson et al., 2007). This POA is formed, as vapors in the combustion exhaust cool down to become supersaturated and condense on existing seed aerosol (Robinson et al., 2010). After emission, POA evaporates with atmospheric dilution since the aerosol mass available for partitioning decreases as the POA is transported away from source regions. Further, diurnal changes in temperature leading to changes in the vapor pressure can also cycle POA between the two phases. Both vapor and particle forms of semi-volatile POA have been shown to photochemically react in the atmosphere to add or remove organic material from the particle-phase (Miracolo et al., 2010) and become more oxygenated (Kroll et al., 2009), although the vapors react much faster. In addition to emissions of "traditional" SOA precursors such as aromatics and long alkanes, all combustion processes are now believed to include emissions of an important additional class of SOA precursors: intermediate-volatility organic compounds (IVOCs) (Jathar et al., 2014). Gas-chromatography mass-spectrometry applications have suggested that while the bulk of the IVOC mass cannot be typically speciated, they are primarily composed of high molecular weight alkanes and aromatics (carbon numbers greater than 12) (Zhao et al., 2014, 2017). Model IVOCs have been shown to form SOA efficiently in chamber experiments (Chan et al., 2009; Lim and Ziemann, 2009; Presto et al., 2010; Tkacik et al., 2012) and have been hypothesized to account for a large fraction of the SOA formed from the photooxidation of motor vehicle exhaust and biomass burning emissions (Jathar et al., 2014; Zhao et al., 2017). The emissions and atmospheric properties (e.g., volatility, reactivity, SOA mass yields) of POA and IVOCs are known (or very likely) to vary by source (e.g., mobile sources versus biomass burning) and hence atmospheric models need to include a source-resolved treatment to accurately predict source contributions to OA and fine particulate matter.

Most commonly used chemical transport models (e.g., CMAQ, CAMx, PMCAMx, WRF-Chem, GEOS-Chem) have been updated to include a semi-volatile and reactive treatment of POA and emissions and SOA formation from IVOCs (Ahmadov et al., 2012; Koo et al., 2014; Murphy and Pandis, 2009; Pye and Seinfeld, 2010). However, their representation in models has been based on very limited and often obsolete data and there are major differences between the implementations in different models. For example, in most models, with a few exceptions (e.g., most recent research version of the OA model in CMAQ developed by Koo et al. (2014)), the gas/particle partitioning of POA was modeled based on measurements performed on a small off-road diesel engine from more than a decade ago (Robinson et al., 2007) and IVOC emissions were based on data gathered from two medium duty diesel vehicles from two decades ago (Schauer et al., 1999). The continued use of these data as inputs for models assumes that not only are these data representative of emissions from modern diesel-powered sources but also that the POA/IVOC properties from diesel sources are similar to those from other sources. Further, the most common schemes to model SOA formation from POA vapors and IVOCs use a single lumped precursor to simulate SOA formation from all sources (e.g., Pye and Seinfeld (2010)) or use an *ad hoc* aging routine that continuously reduces the volatility of the precursor/oxidation products until they partition into the particle phase (Robinson et al., 2007). Most of these schemes have rarely been validated against experimental data, assume that all sources have the same rate and potential to form SOA and, in some cases, ignore fragmentation reactions tied to multigenerational chemistry. As an exception, Hodzic and Jimenez (2011) have used a single lumped precursor to account for all SOA precursors (VOC, IVOC, and SVOC) and the SOA formation from this precursor has been constrained and tested against measurement data. *Ad hoc* aging schemes can overestimate net aerosol mass yields from an SOA precursor and can sometimes overpredict ambient



SOA mass concentrations too, especially over larger regional scales (Dzepina et al., 2009, 2011; Hayes et al., 2015; Jathar et al., 2016). Recently, a host of studies have quantified the volatility of POA emissions from over 100 unique sources and measured SOA formation in more than 100 chamber experiments across six broad source classes: on- and off-road gasoline and diesel sources, wood stoves, and biomass burning (Gordon et al., 2014a, 2014b; Hennigan et al., 2011; May et al., 2013a, 2013b, 2013c, 2014; Tkacik et al., 2017). These data offer a comprehensive set of measurements to inform the source-resolved semi-volatile and reactive behavior of POA and the emissions and SOA formation from IVOCs in atmospheric models.

SOA formation is strongly influenced by the presence of $NO_X$ (Camredon et al., 2007; Chhabra et al., 2010; Loza et al., 2014; Ng et al., 2007b). For most SOA precursors, with the exception of alkanes (Loza et al., 2014) and certain sesquiterpenes (Ng et al., 2007b), environmental chamber data suggest that the reaction chemistry at low $NO_x$, or more precisely low NO, conditions (<2 ppbv) produces more SOA than at high $NO_X$ conditions (>500 ppbv) (Camredon et al., 2007; Chhabra et al., 2010; Loza et al., 2014; Ng et al., 2007b). The consensus seems to be that at low $NO_X$ conditions such as those found in remote continental or marine regions the peroxy radical ($RO_2$) – formed immediately after the reaction of the precursor with the oxidant – combines with the hydroperoxy radical ($HO_2$) or $RO_2$ to form lower volatility hydroperoxides or organic peroxides (Kroll and Seinfeld, 2008). Low NO conditions in remote regions, and in some cases in urban regions that have recently witnessed dramatic reductions in $NO_X$ concentrations, can promote autooxidation reactions to form extremely low-volatility organic compounds (Ehn et al., 2014; Praske et al., 2018). At high $NO_X$, or more precisely high NO, conditions such as those found in urban regions or biomass burning plumes, the $RO_2$ reaction with NO either leads to the formation of alkoxy radicals that can then fragment the carbon backbone, or to the formation of organic nitrates where both reactions result in more volatile products (Kroll and Seinfeld, 2008). Most atmospheric models (e.g., CMAQ, WRF-Chem, GEOS-Chem) have incorporated this knowledge to account for the influence of $NO_x$ on the magnitude, composition, and spatial distribution of SOA.

In the mostly commonly used scheme (e.g., Henze et al. (2008)), $RO_2$ reacts with $HO_2$ to form 'low-NO' SOA or with NO to form 'high-NO' SOA. The $HO_2$:NO ratio determines the branching ratio for $RO_2$ and controls the SOA formed under varying $NO_X$ levels. The SOA yields under the low and high $NO_X$ conditions are parameterized based on chamber data gathered under low and high $NO_x$ conditions respectively. Despite being widely implemented, this scheme has one key limitation that might tend to bias the $NO_X$-dependent predictions of SOA. This scheme relies on an accurate prediction of NO and $HO_2$ to determine the branching ratio for the $RO_2$ radical. Although NO predictions can be validated against routine measurements and most chemical mechanisms seem to predict $NO_X$ (NO+$NO_2$) within a factor of 2, there are very few ambient data to validate model predictions of $HO_2$. For example, as will be shown later, we find that predictions of $HO_2$ concentrations from the use of a typical gas-phase chemical mechanism (SAPRC-11) in a 3D model at Pasadena, CA were almost an order of magnitude lower when compared against measurements at the same site in 2010 (Griffith et al., 2016). In this case, underpredicting $HO_2$ concentrations by an order of magnitude could shift the scheme to produce most of the SOA via the high NO pathway. In contrast, box models that have used the regional atmospheric chemistry mechanism (RACM) have shown good model-measurement comparisons for $HO_2$ concentrations in polluted regions (Griffith et al., 2016; Hofzumahaus et al., 2009). Regardless, gas-phase chemical mechanisms that use the aforementioned scheme need to ensure accurate predictions of $HO_2$ and





NO concentrations to simulate the influence of $NO_X$ on SOA formation.

In this work, we update the organic aerosol model in the UCD/CIT chemical transport model to include a semi-volatile and reactive treatment of POA, emissions and SOA formation from IVOCs, the $NO_X$ influence on SOA formation, and SOA parameterizations for SVOCs and IVOCs that are corrected for vapor wall loss artifacts during chamber experiments. All of these updates are implemented in the statistical oxidation model (SOM) that simulates the oxidation chemistry, thermodynamics, and gas/particle partitioning of OA. Model inputs for POA and IVOCs are based on an interpretation of a comprehensive set of source measurements and resolved broadly by the source type. The $NO_X$ influence on SOA formation is calculated offline based on measured and modeled VOC:$NO_X$ ratios and $NO_X$ concentrations. And finally, the SOA formation from SVOCs and IVOCs is modeled based on recently derived parameterizations that account for vapor wall loss artifacts in chamber experiments. Building on our earlier work (Cappa et al., 2016; Jathar et al., 2015, 2016), these updates within the framework of the SOM have improved the representation of OA in a chemical transport model.

## 2 Methods

### 2.1 Chemical Transport Model

We used the UCD/CIT regional chemical transport model (Kleeman and Cass, 2001) to simulate the emissions, transport, chemistry, and deposition of air pollutants over the state of California at a grid resolution of 24 km and over southern California (see Fig. S1) using a nested 8 km grid from 20th July to 2nd August 2005. The results and analysis were focused on model predictions over Southern California because the region, with approximately 15 million people, is home to one of the most polluted cities in the United States (Los Angeles; ALA (2017)). The time period for simulation was primarily chosen because the model has been previously evaluated for this time period (Jathar et al., 2016) and applied to examine important sources and formation pathways of OA (Cappa et al., 2016; Jathar et al., 2015, 2016, 2017b). The recent literature describes the latest version of the UCD/CIT model but we provide a very brief description of the models and inputs used in this work. Anthropogenic emissions for California were developed using the California Regional PM10/PM2.5 Air Quality Study (CRPAQS) inventory of 2000 but scaled to match conditions in 2005. Wildfire emissions were based on the model FINN (Fire Inventory for National Center for Atmospheric Research) (Wiedinmyer et al., 2011) although they were not found to significantly contribute to OA during the simulated time period (Docherty et al., 2011). Biogenic emissions were based on the model MEGAN (Model of Emissions of Gases and Aerosols from Nature) (Guenther et al., 2006). The Weather Research and Forecasting (WRF) v3.4 model (www.wrf-model.org) was used to produce hourly meteorological fields. National Center for Environmental Protection's NAM (North American Mesoscale) analysis data were used to set the initial and boundary conditions for WRF. The gas- and particle-phase initial and hourly varying boundary conditions were based on the results from the global model MOZART-4/NCEP (Emmons et al., 2010). The gas-phase chemistry was modeled using SAPRC-11 (Carter, 2010).

### 2.2 Organic Aerosol Model

#### 2.2.1 Statistical Oxidation Model (SOM)

In this work, we use the Statistical Oxidation Model (SOM) developed by (Cappa and Wilson, 2012). The SOM is a semi-explicit and parameterizable model that simulates the oxidation chemistry, thermodynamics, and gas/particle





partitioning of OA and its precursors. The SOM has been used to model SOA formation in chamber (Cappa et al., 2013; Cappa and Wilson, 2012; Zhang et al., 2014) and flow reactor (Eluri et al., 2017) experiments. and was recently coupled with SAPRC-11 (gas-phase chemical mechanism) in the UCD/CIT model (Jathar et al., 2015) to investigate the role of chamber-based vapor wall losses (Cappa et al., 2016) and multigenerational aging (Jathar et al., 2016) on the ambient SOA burden. In this work, we used an updated version of the SAPRC-SOM model embedded in the UCD/CIT model that included the POA and IVOC updates described in Section 2.2.2. A detailed description of the mathematical and numerical formulation of the SOM can be found in earlier literature but a brief description of the SOM framework follows. The SOM uses a 2-dimensional carbon-oxygen grid to describe and track the evolution of the gas- and particle-phase organic carbon that is known to yield OA. Each grid cell in the SOM represents an organic species with the molecular formula: $C_XH_{2X+2-Z}O_Z$, where $X=N_C$, and $Z=N_O$. This species is expected to capture the average properties (e.g. volatility, reaction rate constants) of species with the same number of carbon ($N_C$) and oxygen ($N_O$) atoms that are formed from a given SOA precursor. Each species, in the gas and particle phases, is assumed to react with the hydroxyl radical (OH). Operationally, OH is not consumed within the SOM as the chemistry captured in the SOM overlaps with that represented in the gas-phase mechanism (i.e., SAPRC-11). Reactions with the OH radical result in functionalization or fragmentation of the organic species and the distribution of the reaction products is tracked in the carbon-oxygen grid. Six precursor-specific adjustable parameters are assigned for each SOM grid: four parameters that define the molar yields of the four functionalized, oxidized products, one parameter that determines the probability of functionalization or fragmentation and one parameter that describes the relationship between $N_C$, $N_O$ and volatility. In the model, the probability of fragmentation is modeled as a function of the O:C ratio since species with higher O:C ratios have been shown to fragment much more easily than species with lower O:C ratios (Chacon-Madrid and Donahue, 2011). All SOM species properties (e.g., OH reactivity, volatility) are described in terms of $N_C$ and $N_O$.

Seven SOM grids were used to represent SOA formation from nine different precursor classes: (i) long alkanes, (ii) benzene, (iii) high-yield aromatics, (iv) low-yield aromatics, (v) isoprene, (vi) monoterpenes, (vii) sesquiterpenes, (viii) semi-volatile POA (SVOC), and (ix) IVOCs. Classes (i) through (vii) have been included in previous applications of the SOM and we refer the reader to our earlier publications for more details (Cappa et al., 2016; Jathar et al., 2015, 2016). Classes (viii) and (ix) were included in this work for the first time. The SOA formation from monoterpenes and sesquiterpenes (classes vi and vii) was modeled in the same SOM grid since both precursors used the SOM parameter sets for α-pinene. Similarly, the SOA formation from SVOCs and IVOCs was modeled in the same SOM grid and both used the SOM parameter sets for $n$-dodecane; sensitivity simulations were performed using the SOM parameter set for toluene. SOM parameters were determined from fitting the observed SOA volume produced in chamber experiments, with and without accounting for losses of vapors to the chamber walls. We used low and high $NO_x$-specific parameter sets to simulate SOA formation separately under low and high $NO_x$ conditions respectively since the current version of the SOM cannot account for continuous variation in $NO_x$. The SOM parameters used for the nine different classes and seven different grids are listed in Table 1. Parameters for all species except for isoprene were from Cappa et al. (2016). The parameters for isoprene were from Hodzic et al. (2016) that included updates for the reactions rate constants for the first generation products from isoprene photooxidation. Jathar et al. (2016) investigated the influence of oligomerization reactions by allowing irreversible conversion of particle-phase SOM species into a single non-volatile species and found that the oligomerization pathway (as simulated) did





not substantially affect the OA mass concentration in Southern California. Hence, the oligomerization pathway was
not considered in this work. We also did not include the formation of extremely low-volatility organic compounds
from oxidation of SOA precursors such as α-pinene (Ehn et al., 2014) and alkanes (Praske et al., 2018) through
autooxidation pathways, which will very likely be addressed in future versions of the SOM.
*Table 1: SOA precursors and SOM parameters used in this work. VWL=Vapor Wall Loss Corrected, ΔLVP =*
*change in vapor pressure linked to addition of one oxygen atom, $P_{func}$ = molar yields of species that add 1 to 4*
*oxygens per reaction, $m_{frag}$ = exponent influencing the probability of fragmentation.*

| SOA Precursors | SAPRC Species /SOM Grid | SOM Surrogate | VWL | $NO_x$ | ΔLVP | $P_{func}$ | | | | $m_{frag}$ | Reference |
|---|---|---|---|---|---|---|---|---|---|---|---|
| SVOC/IVOC | POA+IVOC | *n*-dodecane/ toluene | No | Low | 1.54 | 0.717 | 0.278 | 0.0028 | 0.0022 | 0.122 | Loza et al. (2014) |
| | | | | High | 1.39 | 0.927 | 0.0101 | 0.018 | 0.0445 | 0.098 | |
| Alkanes | ALK | | Yes | Low | 1.83 | 0.999 | 0.001 | 0.001 | 0.001 | 2 | |
| | | | | High | 1.47 | 0.965 | 0.001 | 0.002 | 0.032 | 0.266 | |
| Benzene | BENZ | benzene | No | Low | 2.01 | 0.769 | 0.001 | 0.0505 | 0.180 | 2.010 | Ng et al. (2007a) |
| | | | | High | 1.7 | 0.079 | 0.001 | 0.919 | 0.001 | 0.535 | |
| | | | Yes | Low | 1.97 | 0.637 | 0.001 | 0.002 | 0.360 | 0.0807 | |
| | | | | High | 1.53 | 0.008 | 0.001 | 0.991 | 0.001 | 0.824 | |
| High-yield aromatics | ARO1 | toluene | No | Low | 1.84 | 0.561 | 0.001 | 0.001 | 0.438 | 0.010 | Zhang et al. (2014) |
| | | | | High | 1.24 | 0.003 | 0.001 | 0.001 | 1.010 | 0.222 | |
| | | | Yes | Low | 1.77 | 0.185 | 0.001 | 0.002 | 0.812 | 1.31 | |
| | | | | High | 1.42 | 0.856 | 0.001 | 0.002 | 0.141 | 4.61 | |
| Low-yield aromatics | ARO2 | *m*-xylene | No | Low | 1.76 | 0.735 | 0.001 | 0.002 | 0.262 | 0.010 | Ng et al. (2007a) |
| | | | | High | 1.68 | 0.936 | 0.001 | 0.002 | 0.061 | 0.010 | |
| | | | Yes | Low | 2.05 | 0.102 | 0.001 | 0.878 | 0.019 | 1.08 | |
| | | | | High | 1.46 | 0.001 | 0.001 | 0.942 | 0.056 | 0.0671 | |
| Isoprene | ISOP | isoprene | No | Low | 2.26 | 0.973 | 0.001 | 0.001 | 0.026 | 0.010 | Chhabra et al. (2011); Hodzic et al. (2016) |
| | | | | High | 1.94 | 0.952 | 0.001 | 0.030 | 0.016 | 0.063 | |
| | | | Yes | Low | 2.25 | 0.1646 | 0.5164 | 0.3012 | 0.0179 | 0.0244 | |
| | | | | High | 1.93 | 0.988 | 0.0002 | 0.0116 | 0.0009 | 0.51 | |
| Monoterpenes /Sesquiterpenes | TRP | α-pinene | No | Low | 1.87 | 0.001 | 0.869 | 0.078 | 0.053 | 0.010 | Chhabra et al. (2011) |
| | | | | High | 1.62 | 0.068 | 0.633 | 0.275 | 0.024 | 0.035 | |
| | | | Yes | Low | 1.97 | 0.419 | 0.426 | 0.140 | 0.014 | 0.305 | |
| | | | | High | 1.91 | 0.500 | 0.422 | 0.070 | 0.008 | 0.16 | |

**2.2.2 Model Inputs**
*Semi-Volatile and Reactive POA (SVOC).* POA from gasoline, diesel, biomass burning, and food cooking sources was
treated as semi-volatile and reactive. POA from all other sources (e.g., marine, dust) was assumed to be non-volatile in
all simulations except one where we explored the sensitivity in model predictions to this assumption (see Section 2.3
for more details). Semi-volatile POA was modeled by distributing POA emissions from the emissions inventory in the
SOM grid as hydrocarbon species modeled as linear alkanes, i.e. as species with no oxygen (i.e., $C_X H_Y$). The
hydrocarbon/linear alkane distribution in the SOM grid was determined by refitting the volatility distributions
published by May and coworkers (May et al., 2013a, 2013b, 2013c) such that the hydrocarbon distribution reproduced
the observed gas/particle partitioning behavior; the hydrocarbon distributions are listed in Table S1. We assumed all
on- and off-road gasoline exhaust POA to have the same hydrocarbon/linear alke distribution as the volatility



distribution determined by May et al. (2013a) from data for 67 light-duty gasoline vehicles. Based on tests performed on eight light-duty gasoline vehicles, Kuwayama et al. (2015) found that the POA volatility for their vehicles was consistent with that determined by (May et al., 2013a) for about half the vehicles but substantially lower for the other half. They hypothesized that the lower POA volatility could be attributed to fuel oxidation products. The findings of Kuwayama et al. (2015) suggest that the volatility distribution used in this work may overestimate the evaporation of POA with dilution. We assumed all on- and off-road diesel exhaust POA to have the same hydrocarbon/linear alkane distribution as the volatility distribution determined by May et al. (2013b) from data for two medium-duty diesel trucks, three heavy-duty diesel trucks, and a single off-road diesel engine. We assumed residential wood combustion and wildfires to have the same hydrocarbon/linear alkane distribution as the volatility distribution determined by May et al. (2013c) from a selection of fifteen different fuels. We assumed food cooking to have the same hydrocarbon/linear alkane distribution as that for wildfires although recent work suggests that food cooking OA may be significantly less volatile than wildfire OA (Louvaris et al., 2017; Woody et al., 2016). Hence, model predictions in this work probably underestimate the POA mass concentrations and overestimate the SVOC emissions from food cooking sources. This work, similar to the most recent implementation in the Community Multiscale Air Quality (CMAQ) model (Koo et al., 2014; Woody et al., 2016), included a source-resolved treatment of semi-volatile POA that was tied to a comprehensive set of source measurements.

The reactive behavior of POA was modeled by assuming that the POA vapors (i.e. SVOCs) (represented as a hydrocarbon distribution) and their products participated in gas-phase oxidation and formed SOA similar to linear alkanes and utilized the SOM parameter set for $n$-dodecane. The surrogate, in this case $n$-dodecane, only informs the multi-generational oxidation chemistry of the precursor and the actual compound of interest (e.g., a $C_{15}$ linear alkane) can have a different SOA mass yield than that of $n$-dodecane. The reaction rate constants with OH for the parent hydrocarbons were assumed to be similar to the carbon-equivalent linear alkane. The equivalence to linear alkanes while not perfect was probably a good assumption for gasoline and diesel sources since alkanes account for a substantial fraction of gasoline and diesel fuel (Gentner et al., 2012) and lubricating oil (Caravaggio et al., 2007) are a dominant organic class in both gas- and particle-phase emissions from mobile sources (Brandenberger et al., 2005; Hays et al., 2017; Schauer et al., 1999, 2002b)(Worton et al., 2014). However, alkanes do not make up a significant fraction of the gas- and particle-phase emissions from biomass burning (Hatch et al., 2015; Schauer et al., 2001; Stockwell et al., 2015) or food cooking (Schauer et al., 2002a) and hence it is unclear whether linear alkanes are good surrogates to model the oxidation of SVOCs from these sources. To test the sensitivity of the model predictions to the surrogate used to model SOA formation from SVOCs, we ran sensitivity simulations where we modeled the SVOCs as a mixture of aromatic compounds using the SOM parameter set for toluene (see rationale in Section 2.4).

*Intermediate-Volatility Organic Compounds.* We included IVOC emissions from gasoline, diesel, and biomass burning. We assumed none of the other sources emitted IVOCs for all simulations except one where we explored the sensitivity in model predictions to this assumption (see Section 2.4 for more details). The IVOC emissions estimates and their potential to form SOA was based on the work of Jathar et al. (2014). In Jathar et al. (2014), IVOC emissions, defined as the sum of all unspeciated compounds, were determined as a mass fraction of the total non-methane organic gas (NMOG) emissions for three different source categories: gasoline vehicles, diesel vehicles, and biomass burning. Based on that work, IVOCs were assumed to be 25% of the NMOG emissions for on- and off-road gasoline exhaust,





20% of the NMOG emissions for on- and off-road diesel exhaust, and 7% of the NMOG emissions for residential wood combustion and wildfires. No IVOCs were considered for the food cooking source but recent work suggests that they might play a role in influencing the OA evolution from a multitude of food cooking sources (Kaltsonoudis et al., 2017; Liu et al., 2017). We assumed the NMOG emissions in the emissions inventory to account for most of the gas-phase organic compound mass that included the IVOCs and hence the addition of IVOC emissions meant that the non-IVOC emissions had to be reduced to conserve total NMOG mass. Recent literature suggests that IVOCs could be lost to walls of the sampling hardware (e.g., tubing, bags) (Pagonis et al., 2017) and therefore would be excluded in the NMOG measurement. Our assumption should result in conservative estimates for the influence of IVOC emissions on SOA formation. Following Jathar et al. (2014), the IVOCs were modeled as a $C_{13}$ hydrocarbon for those from on- and off-road gasoline sources and as a $C_{15}$ hydrocarbon for those from on- and off-road diesel sources and biomass burning. The oxidation of the IVOC hydrocarbons and their reaction products and the subsequent SOA formation was modeled assuming equivalence to a linear alkane and used the SOM parameter set for *n*-dodecane. The equivalent linear alkane to model SOA formation from IVOCs in Jathar et al. (2014) was based on fitting the SOA formation observed in chamber experiments and hence the choice of the hydrocarbon in this work was experimentally constrained. We also investigated the sensitivity in model predictions to the use of an aromatic compound (i.e., toluene) as a surrogate instead of an alkane (i.e., *n*-dodecane) to model SOA formation from IVOCs (see rationale in Section 2.4).

Recently, Zhao and coworkers (Zhao et al., 2015, 2016) used thermal desorption gas-chromatography mass spectrometry (TD-GC-MS) to measure IVOC emissions in gasoline and diesel exhaust and speciated/classified the IVOCs as a mixture of linear, branched, and cyclic compounds resolved by carbon number. We should note that Zhao et al. (2015, 2016) defined IVOCs as the sum of speciated and unspeciated hydrocarbons roughly larger than a $C_{12}$ alkane, which was different from the definition adopted by Jathar et al. (2014). In their first paper, Zhao et al. (2015) found IVOCs to be about 60% of the NMOG mass emissions for tailpipe exhaust from older diesel vehicles/engines (ones without particle filters or oxidation/reduction catalysts). In this work we used an IVOC:NMOG ratio of 0.2 and likely underestimated IVOC emissions from diesel sources by a factor of 2.5. Zhao et al. (2015) concluded that the effective IVOC yield based on their speciation was comparable to the yield of the $C_{15}$ linear alkane used in this work but the application of that yield over-predicted the chamber SOA data from Gordon et al. (2014a) by a factor of 1.8; virtually all of the SOA predicted by Zhao et al. (2016) was from the oxidation of IVOCs. If one assumed that the effects from lower IVOC emissions (factor of 2.5) were roughly balanced by the use of higher SOA yields (factor of 1.8), then the SOA formation from diesel sources was probably well represented in our work.

In their second paper, Zhao et al. (2016) found the IVOCs to be only about 4% of the NMOG mass emissions in gasoline exhaust but we used an IVOC:NMOG ratio of 0.25 in this work. This suggests that we may be overestimating the gasoline exhaust IVOC emissions by approximately a factor of six in this work. Based on the speciation performed, Zhao et al. (2016) estimated that the IVOCs collectively had an SOA yield between 19 and 24% at an OA mass concentration of 9 $\mu g\ m^{-3}$ (9 $\mu g\ m^{-3}$ was the average end-of-experiment concentration in the chamber experiments of Gordon et al. (2014a)), which was slightly more than twice the SOA yield for a $C_{13}$ linear alkane (7-12%) – used to model gasoline IVOCs in this work – at the same OA mass concentration. However, application of the Zhao et al. (2016) SOA yields for IVOCs underpredicted the observed chamber SOA formation for newer gasoline



vehicles by a factor of ~2. Since IVOC oxidation accounted for slightly less than half of the SOA formed (with the
other half coming from single-ring aromatics), the IVOC SOA yields in Zhao et al. (2016) would need to be tripled to
explain the chamber SOA measurements. If we assumed that the effects from higher IVOC emissions (factor of 6)
were approximately balanced by the use of lower SOA yields (factor of 2×3=6), then the SOA formation from
gasoline sources in this work was probably well represented in our work.
To summarize, the IVOC emissions estimates and the surrogates used to model SOA formation from IVOCs from
gasoline and diesel sources in this work, while different from those suggested in Zhao et al. (2015, 2016), are still
consistent with the SOA measurements made by Gordon et al. (2014a, 2014b).

### 2.2.3 Modeling the $NO_X$ Dependence on SOA Formation

Previous applications of the SOM have simulated SOA under low and high $NO_X$ conditions separately since the SOM,
in its current form, cannot model the continuous evolution of SOA under varying $NO_X$ conditions using the local
$NO/HO_2$. Predictions from either of these simulations (Jathar et al., 2016) or the average of these simulations (Cappa
et al., 2016) likely do not accurately characterize the evolution or spatial distribution of SOA since $NO_X$
concentrations exhibit strong spatial variability with higher concentrations in urban (e.g., traffic) and source (e.g.,
wildfires) regions. For example, since most precursors have higher SOA yields under low $NO_X$ conditions than under
high $NO_x$ conditions, the use of an average is expected to overestimate SOA in high-$NO_X$ urban areas and
underestimate SOA in low-$NO_X$ rural/remote continental areas.
In this work, we used two different offline techniques to account for the influence of $NO_X$ on SOA formation. For
both methods, we assumed that the 3D model predictions based on the low and high $NO_X$ SOA parameterizations
bounded the minimum and maximum ambient SOA mass concentrations. (Xu et al., 2015) found that the SOA
formation from isoprene photooxidation was maximized at intermediate $NO_X$ levels with lower values at the extreme
$NO_X$ levels, suggesting that our assumption may not necessarily hold for all precursor species. In the first method, we
used the VOC:$NO_X$ ratios from the low and high $NO_X$ chamber experiments as our bounds and used the 3D model
predicted VOC:$NO_X$ ratio to interpolate between the minimum and maximum SOA mass concentrations predicted
from the low and high $NO_X$ simulations. In the second method, we used $NO_X$ concentrations from the low and high
$NO_X$ chamber experiments and the 3D model predictions to perform the interpolation. For each method, we performed
the interpolation on the SOA mass concentrations assuming a linear or logarithmic dependence on the VOC:$NO_X$
ratios and $NO_X$ concentrations. The VOC:$NO_X$ ratio and the $NO_X$ concentration served as an approximate surrogate
for the $HO_2$:NO ratio used in most atmospheric models to simulate the $NO_X$-dependent SOA formation. The $NO_X$-
adjusted SOA concentrations ($SOA_{eff}$) from each precursor at each grid cell were calculated from model predictions
from the low and high $NO_X$ simulations using the following equations:
$SOA_{eff} = SOA_{high\,NO_X} + \frac{SOA_{low\,NO_X} - SOA_{high\,NO_X}}{(VOC:NO_x)_{low\,NO_X} - (VOC:NO_x)_{high\,NO_X}} \times ((VOC:NO_x)_{model} - (VOC:NO_x)_{high\,NO_X})$ - (1)
$SOA_{eff} = SOA_{high\,NO_X} + \frac{SOA_{low\,NO_X} - SOA_{high\,NO_X}}{log(VOC:NO_x)_{low\,NO_X} - log(VOC:NO_x)_{high\,NO_X}} \times (log(VOC:NO_x)_{model} -$
$log(VOC:NO_x)_{high\,NO_X})$ - (2)





$$SOA_{eff} = SOA_{low\,NO_x} - \frac{SOA_{low\,NO_x} - SOA_{high\,NO_x}}{(NO_x)_{high\,NO_x} - (NO_x)_{low\,NO_x}} \times ((NO_X)_{model} - (NO_X)_{low\,NO_x}) \text{ - (3)}$$
$$SOA_{eff} = SOA_{low\,NO_x} - \frac{SOA_{low\,NO_x} - SOA_{high\,NO_x}}{log(NO_x)_{high\,NO_x} - log(NO_x)_{low\,NO_x}} \times (log(NO_x)_{model} - log(NO_x)_{low\,NO_x}) \text{ - (4)}$$
where $SOA_{low\,NO_X}$ and $SOA_{high\,NO_X}$ are model predictions of SOA from using the low and high $NO_X$
parameterizations respectively, $(VOC:NO_X)_{low\,NO_X}$ and $(VOC:NO_X)_{high\,NO_X}$ are the initial VOC:NO$_X$ ratios from the
chamber experiments used to develop the low and high NO$_X$ SOA parameterizations, $(VOC:NO_X)_{model}$ is the model
predicted VOC:NO$_X$ ratio in the model grid cell, $(NO_X)_{low\,NO_X}$ and $(NO_X)_{high\,NO_X}$ are the NO$_X$ concentrations from
the chamber experiments used to develop the low and high NO$_X$ parameterizations, and $(NO_X)_{model}$ is the model
predicted NO$_X$ concentration in the model grid cell. Equations (1) and (3) assume linear dependence while equations
(2) and (4) assume logarithmic dependence. For the $(VOC:NO_x)_{model}$ ratio, the VOC is the sum of all organic species
tracked in the SAPRC-11 gas-phase chemical mechanism. NO$_X$ is the sum of NO and NO$_2$. The $(VOC:NO_x)$ ratios
and the $NO_X$ concentrations from the chamber experiments used in the equations were gathered directly from the
primary references and are listed in Table 2. When the $(VOC:NO_X)_{model}$ or $(NO_X)_{model}$ values were lower or higher
than the chamber values in Table 2, the SOA formation was set to model predictions from the bounding simulations.
Table 2: Low and high VOC:NO$_x$ ratios in ppb ppb$^{-1}$ from chamber experiments used to model the influence of NO$_X$ on
SOA formation.

| SOM surrogate | $(VOC:NO_x)_{low\,NO_x}$ | $(NO_X)_{low\,NO_x}$ | $(VOC:NO_x)_{high\,NO_x}$ | $(NO_X)_{high\,NO_x}$ | Reference |
|---|---|---|---|---|---|
| *n*-dodecane | 17.0 | <2 ppbv | 0.09 | 343 | Loza et al. (2014) |
| benzene | 207 | <2 ppbv | 1.98 | 169 | Ng et al. (2007a) |
| toluene | 46.3[*] | <0.8 ppbv | 0.76[*] | 50 | Zhang et al. (2014) |
| *m*-xylene | 12.1[#] | <2 ppbv | 0.10 | 943 | Ng et al. (2007a) |
| isoprene | 24.5 | <2 ppbv | 0.29 | 937 | Chhabra et al. (2010) |
| α-pinene | 33.1 | <2 ppbv | 0.05 | 844 | Chhabra et al. (2010) |

[&] minimum VOC:NO$_x$ ratios since these assume a NO$_X$ concentration of 0.8 ppbv in the chamber
[*] average of six experiments performed by Zhang et al. (2014)
[#] average of two experiments performed by Ng et al. (2007a)
We acknowledge that this approach to modeling the NO$_X$ influence on SOA formation is limited and is sensitive to the
following assumptions: (i) the VOC:NO$_X$ ratio plus NO$_X$ concentration is a good proxy to model the HO$_2$:NO ratio
and the branching between low and high NO$_X$ SOA formation, (ii) the low and high NO$_X$ chamber experiments for a
particular precursor bound the minimum and maximum SOA formed, (iii) the SOA response between the low and
high NO$_X$ levels varies linearly or logarithmically with VOC:NO$_X$ ratios/NO$_X$ concentrations, and (iv) the model
predicted VOC concentrations at each grid cell, summed across a mixture of organic compounds, are analogous to the
initial VOC concentrations from the chamber experiment to calculate VOC:NO$_X$ ratios. There are few experimental
data to test these assumptions and these need to be investigated in future work. In addition to modeling the influence
of NO$_X$ on ambient SOA concentrations, this approach allowed us to explore the influence of reductions in NO$_X$
emissions and concentrations on ambient OA concentrations in the future.
**2.3 Simulations**
Table 3: Names and descriptions of the simulations performed in this work





| No. | Name | Semi-volatile & Reactive POA (SVOC) | IVOC | Vapor Wall Losses for SVOC and IVOC | Additional Details |
|---|---|---|---|---|---|
| 1 | Traditional | No | No | No | Same as model of Cappa et al. (2016) |
| 2 | SVOC | Yes | No | No | - |
| 3 | IVOC | Yes | Yes | No | - |
| 4 | Base | Yes | Yes | Yes | Base case model used in this work |
| 5 | - SVOC$_{max}$* | | | | SVOCs modeled as per diesel parameterization |
| 6 | - IVOC$_{max}$* | | | | IVOCs modeled as per diesel parameterization |
| 7 | - No-Aging* | | | | No multi-generational aging |
| 8 | - VOC$_{spec}$* | | | | VOC speciation from May et al. (2014) |
| 9 | - Aromatic* | | | | S/IVOCs modeled using the toluene parameterization |

*Same as the Base simulation but with differences noted in the 'Additional Details' section.

The Base simulation – representing our most comprehensive simulation – included the updates described in Section 2.2.2; a source-resolved semi-volatile and reactive treatment of POA, source-resolved SOA formation from SVOCs and IVOCs, and correction of the subsequent SOA formation for vapor wall losses in chambers. The Base simulation included sub-simulations at two resolutions (24 km and 8 km) with two $NO_X$ parameterizations (low and high $NO_X$).

Additional simulations were designed and performed with two objectives in mind: (i) to examine the influence of each update included in this work and (ii) to test the sensitivity in model predictions to uncertainties inherent in the updates and other model inputs. A set of four simulations were performed to systematically study the influence of model updates. These included the following simulations where only one update (as underlined) was made over the previous configuration: (1) Traditional – Non-volatile POA, no IVOCs, SOA from VOCs, and no correction for chamber vapor wall losses, (2) SVOC – Semi-volatile POA, no IVOCs, SOA from SVOCs and VOCs, and no correction for chamber vapor wall losses, (3) IVOC – Semi-volatile POA, IVOCs, SOA from SVOCs, IVOCs, and VOCs, and no correction for chamber vapor wall losses, and (4) Base – Semi-volatile POA, IVOCs, SOA from SVOCs, IVOCs, and VOCs, and correction for chamber vapor wall losses. Successive differences in model predictions between the Traditional, SVOC, IVOC, and Base simulations were used to systematically examine the influence of the semi-volatile and reactive POA, IVOCs, and chamber vapor wall losses respectively.

A set of five simulations were performed to study uncertainties in model inputs. The SVOC$_{max}$ simulation (5) assumed that POA from all sources (all POA except marine POA) was semi-volatile and modeled using the volatility distribution for diesel exhaust POA. Diesel POA was chosen since it was the most volatile of the volatility distributions used in this work. This simulation bounded the maximum loss in POA mass to evaporation. The IVOC$_{max}$ (6) simulation assumed that all sources (combustion and non-combustion except biogenic sources) emitted IVOCs, which were estimated using an IVOC:NMOG ratio of 0.2 and allowed to form SOA equivalent to a $C_{15}$ alkane. This simulation provided an upper bound estimate to the contribution of IVOCs to ambient SOA although the IVOC emissions and their potential to form SOA could be even higher than that assumed here. The No-Aging (7) simulation assumed no multi-generational aging or in other words, the emitted precursor was allowed to react with OH and form four functionalized products with no further oxidation. This simulation investigated the influence of multi-generational aging on ambient SOA. The VOC$_{spec}$ (8) simulation updated the VOC speciation for on- and off-road





gasoline and diesel vehicles based on a comprehensive set of measurements performed on an in-use fleet (May et al., 2013a, 2013b). This simulation examined the influence of updated emissions profiles on the non-IVOC contribution to SOA. Finally, the Aromatic (9) simulation assumed that the oxidation of SVOCs and IVOCs to form SOA was modeled using toluene. There were two reasons for choosing toluene. First, both mono- and poly-cyclic aromatic compounds are found in gasoline and diesel fuel (Gentner et al., 2012) and in tailpipe emissions from mobile sources (Zhao et al., 2015, 2016), and oxygenated aromatic compounds such as phenols, guaiacols, and syringols are found in biomass burning emissions (Schauer et al., 2001; Stockwell et al., 2015). Second, aromatic compounds, similar to alkanes, have been studied in detail for their potential to form SOA and are recognized to form more SOA than linear alkanes for the same carbon number. This simulation provided an upper bound estimate for SOA formation from the oxidation of SVOCs and IVOCs.

The UCD/CIT model was run on the High Performance Computing cluster run by Engineering Network Services at Colorado State University. Although the number of cores varied based on availability, on average each simulation used 96 cores and required 5 days to execute 19 simulated days. Since each set included four sub-simulations, each simulation required ~5 days and all simulations in this work required ~180 days of computational time.

## 2.4 Measurements for Model Evaluation

Model predictions were evaluated against gas-phase measurements of SOA precursors and particle-phase measurements of OA mass concentrations and composition. Here, we briefly describe the primary measurement data and any post-processing of the data we performed prior to undertaking the model evaluation.

Gas-phase measurements of SOA precursors were from two different sources. The first source was routine daily-averaged measurements of single-ring aromatics made by the South Coast Air Quality Management District (SCAQMD, 2017) in southern California at three different sites: North Los Angeles, Riverside, and Long Beach. While measurement data were available at three other sites, data were not available for 2005, our modeled year and hence not included. These gas-chromatography-based measurements were available every twelfth day and included the following aromatic species: benzene, toluene, *o/m/p*-xylene, ethyl-benzene, and styrene. Since there was little overlap between the modeled episode (14 day period over July-August) and available aromatic data, the measurement data were averaged over a three month period in the summer (May 15th to September 15th) and then compared to the episode-averaged model predictions. The second source was gas-chromatography mass-spectrometry measurements of single-ring aromatics (Borbon et al., 2013) and IVOCs (Zhao et al., 2014) made at the Pasadena ground site in the months of May and June of 2010 as part of the CalNex campaign. The single-ring aromatics were measured every hour and included the following species: benzene, toluene, *o/m/p*-xylene, ethyl-benzene, and styrene. The IVOCs were measured every three hours and included most of the reduced and oxidized organic species with a carbon number larger than 12. Since these measurements were from a different time period, we compared campaign-averaged measurements against episode-averaged model predictions.

Particle-phase measurements were from two different sources as well. The first source was routine daily-integrated measurements of organic carbon (OC) in southern California from four sites in the Chemical Speciation Network (CSN; Central Los Angeles, Riverside, Simi Valley, and Escondido) and six sites in the Interagency Monitoring of





Protected Visual Environments (IMPROVE) network (San Rafael, Rubidoux-Riverside, San Gorgonio Wilderness, Joshua Tree NP, Agua Tibia, and San Gabriel). The CSN is a network of ~50 urban measurement sites across the United States where pollutant concentrations are typically higher, more variable, and representative of local sources and measurements are made once every three days. The IMPROVE is a network of ~200 rural/remote continental sites typically located in national parks across the United States where pollutant concentrations are lower, less variable, and representative of regional influences and measurements are made once every three days. Over the 14 day episode modeled in this work, three measurements from the CSN and five measurements from the IMPROVE network were available for comparison. We used an organic aerosol to organic carbon ratio (OA:OC) of 1.6 to calculate OA at the CSN sites (Docherty et al. (2011) measured an OA:OC ratio of 1.77 during the SOAR-1 campaign, after correction with the updated calibration of Canagaratna et al. (2015)) and a ratio of 2.1 to calculate OA at the IMPROVE sites (Turpin and Lim, 2001). The CSN data are artifact corrected but we subtracted 0.5 μg m$^{-3}$ from the calculated OA mass concentrations to blank correct the data (Subramanian et al., 2004). The IMPROVE data are both blank and artifact corrected. We note that a negative evaporation artifact has been reported for at IMPROVE sites in the southeast US (Kim et al., 2015) but it is not known whether such an artifact may be present in this region and no correction has been made. The second source was particle measurements made at the ground site in Riverside as part of the SOAR-1 campaign during the summer of 2005 (Docherty et al., 2008, 2011). These measurements included hourly-averaged mass concentrations and elemental ratios of H:C and O:C for OA, and estimates of the POA-SOA split based on results from a positive matrix factorization analysis.

## 3 Results

### 3.1 POA and SOA Precursor Emissions

Gas- and particle-phase emissions of organic compounds in the 8 km southern California domain, averaged over the 14-day episode, are shown in Figure 1. The 8 km domain, shown in Figure S1, includes the entire Los Angeles metropolitan statistical area, parts of the Pacific Ocean, and forested areas surrounding the urban area. The emissions are color-coded by source type and include all species that contribute to direct emissions and atmospheric formation of OA. These do not include emissions of marine POA since those were calculated inline in the UCD/CIT model. Since the POA repartitioned between the gas and particle phases after emission, POA was split into POA and SVOC that represented the particle and gas portions of POA partitioned at an urban OA mass concentration of 9 μg m$^{-3}$. We chose 9 μg m$^{-3}$ to partition POA because the campaign-averaged OA mass concentration at Riverside during SOAR-1 was 9 μg m$^{-3}$. If one discounts the POA emissions in the 'Other' category (which is mostly made of road, agricultural, and construction dust), the re-partitioning resulted in about 60% of the POA emitted to evaporate as SVOC vapors; these vapors will oxidize in the atmosphere to form SOA. As noted earlier, a relatively more volatile treatment compared to that described in the recent literature suggests that we may have overestimated the POA evaporation from food cooking sources. Mobile sources accounted for 20% of the POA and 35% of the SVOC vapors and competed with food cooking as an important source of primary emissions and one which accounted for 15% of the POA and 44% of the SVOC vapors. IVOC, long alkane, and aromatic emissions were roughly on the same order of magnitude but taken together were approximately an order of magnitude larger than the POA emissions. This suggests that even at low SOA mass yields (say <10%), the OA formed from the oxidation of these precursors could quickly exceed direct emissions of POA.



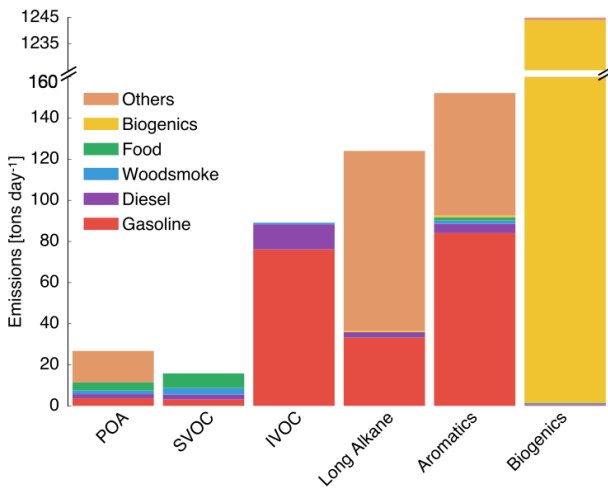

*Figure 1: Episode-averaged gas- and particle-phase organic emissions in tons per day over the 8 km southern*

*California domain resolved by source. POA and SVOC represent the particle- and gas-phase emissions partitioned to*

*an OA mass concentration of 9 μg m$^{-3}$. SVOC, IVOC, long alkanes, aromatics, and biogenics represent gas-phase*

*emissions of precursor species that are modeled to form SOA.*

Emissions of total IVOCs were slightly lower than those for long alkanes (by ~30%) and aromatics (by ~40%) but a
factor of 2 higher than the sum of POA and SVOCs. Previously, IVOC emissions have been estimated by scaling
POA emissions by a factor of 1.5 to 4 derived from gas/particle partitioning calculations (Dzepina et al., 2009;
Shrivastava et al., 2008) and from atmospheric measurements (Ma et al., 2017). While our estimate for IVOC
emissions are within the previously used range, our estimates were informed by a broader suite of source
measurements, which will help reduce the uncertainty in IVOC emissions and related SOA formation in atmospheric
models. IVOC emissions from mobile sources were similar to aromatic emissions but twice the long alkane emissions.
Hence, we anticipated IVOCs to contribute meaningfully to the anthropogenic SOA burden. We note that in this work
we only considered IVOC emissions from combustion sources but recent work suggests that volatile chemical
products present in sources such as pesticides, coatings, cleaning agents, and personal care products may be a large
source of SVOCs and IVOCs in urban environments (McDonald et al., 2018).
Mobile sources – dominated by gasoline use – accounted for a much larger fraction of the anthropogenic SOA
precursors (85% of IVOCs, 27% of long alkanes, and 55% of aromatics) in this study. Hence, mobile source
regulation on precursor emissions from gasoline vehicles (e.g., limits on emissions of unburned hydrocarbons) has and
could have a much larger influence on controlling ambient OA than regulating direct emissions of POA, although this
ultimately depends on the extent of conversion of these species to SOA. Finally, biogenic precursor emissions of
isoprene, monoterpenes, and sesquiterpenes were about a factor of three higher than the combined emissions of
SVOCs, IVOCs, long alkanes, and aromatics and will continue to be an important source of SOA in southern
California. However, their impact on urban OA/SOA will be smaller since these emissions are primarily limited to
regions outside the urban areas.





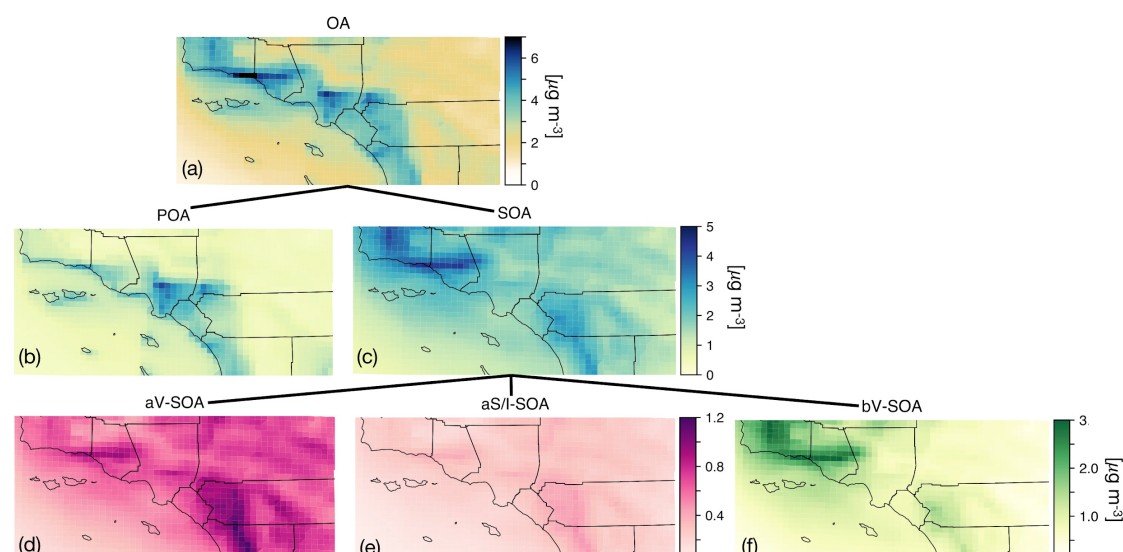

*Figure 2: 14-day averaged model predictions of mass concentrations for OA, POA, SOA, aV-SOA, aS/I-SOA, and bV-*
*SOA in µg m$^{-3}$ over the southern California domain from the Base simulation.*
**3.2 Spatial Distribution of OA Concentrations and Bulk Composition**
In Figure 2 we plot predictions of the 14-day averaged mass concentrations for OA, POA, SOA, and contributions
from three lumped SOA precursors (long alkanes and aromatics, SVOC and IVOCs, and biogenic VOCs) from the
Base case simulation. We used the terminology developed by Murphy et al. (2014) to describe the SOA from the
different sources. To reiterate, the Base case simulation included a semi-volatile treatment of POA, SOA formation
from oxidation of SVOCs, IVOCs, and VOCs, multi-generational aging, and SOA parameterizations that accounted
for the influence of chamber vapor wall losses. The mass concentrations in Figure 2 account for SOA formation under
varying NO$_X$ levels as per equation 2 (logarithmic dependence on the VOC:NO$_X$ ratio). We chose equation 2 because
it produced the highest SOA mass concentrations and presented an upper bound on SOA formation.
The highest OA mass concentrations were found in three general regions: the densely-populated Los Angeles-Orange-
Riverside County region likely attributed to heavy transportation emissions, along the coast as a result of sea spray
emissions, and in biogenic VOC dominated areas. In central Los Angeles, OA accounted for 38% of the modeled non-
refractory PM$_{2.5}$ mass concentration with 20, 25, and 18% contributions from sulfate, nitrate, and ammonium aerosol.
A sensitivity simulation that turned emissions of marine POA off suggested that the marine POA mass concentrations
in central Los Angeles were ~0.9 µg m$^{-3}$, which were considerably higher than the coastal measurements made during
CalNex in 2010 (Hayes et al., 2013). Measured mass concentrations of POA over the open ocean west of California
were ~0.2 µg m$^{-3}$ during CalNex in 2010 and it was expected that these mass concentrations would be substantially
lower by the time they were transported to central Los Angeles. Sea spray emissions in the UCD/CIT model are based
on the parameterization of Gong et al. (2003) and may need to be revisited in the future.





The broader spatial trends of OA, POA, and SOA were in line with results from earlier chemical transport model
studies that have treated POA as semi-volatile and modeled SOA formation from SVOCs and IVOCs (Ahmadov et
al., 2012; Jathar et al., 2017a; Koo et al., 2014; Robinson et al., 2007; Tsimpidi et al., 2010). POA mass
concentrations were highest in upwind (e.g., 3.4 µg m$^{-3}$ in central Los Angeles) and lower in downwind (e.g., 2.7 µg
m$^{-3}$ in Riverside) locations as the POA emissions that were transported away from the source region evaporated with
dilution. SOA mass concentrations, compared to POA, had a more regional presence with higher concentrations either
downwind of the source regions (e.g., 2.4 µg m$^{-3}$ in Riverside versus 2.2 µg m$^{-3}$ in central Los Angeles) or in regions
with high emissions of biogenic VOCs (e.g., 2.5 µg m$^{-3}$ inside the Los Padres National Forest). To assess the relative
contribution of POA and SOA to total OA, we plot the POA to SOA ratio in Figure S2, which suggests a POA-SOA
ratio of ~1.6 in near-source regions and lower elsewhere, e.g., ~0.4, 0.8, and 1.2 in representative marine, biogenic-
dominated, and urban downwind regions. These POA-SOA splits qualitatively aligned with the hydrocarbon-like and
oxygenated organic aerosol (HOA and OOA) splits estimated in aerosol mass spectrometer datasets in urban locations
worldwide (Jimenez et al., 2009; Zhang et al., 2007). However, we predict POA/SOA ~1 for Riverside during SOAR-
1, compared to a measured ratio of ~0.25 (Docherty et al., 2008), which indicates that SOA may still be
underestimated in the model  A comparison of the OA composition predictions with the aerosol mass spectrometer
measurements is described in Section 4.
Panels (d) through (f) show contributions of three distinct SOA precursor classes to total SOA. Alkane and aromatic
VOCs – included as SOA precursors in most atmospheric models – appeared to contribute a maximum of 1.2 µg m$^{-3}$
of what we refer to as aV-SOA downwind of the source region. The majority of this aV-SOA (75% ) originated from
aromatic precursors implying that alkane VOCs are unlikely to contribute much to the anthropogenic SOA or total OA
burden in urban areas, consistent with our earlier work (Cappa et al., 2016; Jathar et al., 2016). We note that emissions
inventories typically only include alkane species with carbon numbers less than 12 (Pye and Pouliot, 2012) and longer
alkanes with carbon numbers larger than 12 are included as part of the POA, SVOC, and IVOC emissions. Together
aS-SOA and aI-SOA mass concentrations exhibited a similar spatial pattern over the domain but were substantially
lower than the aV-SOA mass concentrations – reaching a maximum of only 0.5 µg m$^{-3}$. The lower aS-SOA and aI-
SOA mass concentrations were somewhat contrary to earlier work that has argued that SVOCs and IVOCs are an
equal or dominant precursor of anthropogenic SOA when compared to aV-SOA, especially in urban areas (Jathar et
al., 2014, 2017a; Woody et al., 2016). The reason for these lower concentrations can be partially attributed to the
precursor-dependent influence of accounting for vapor wall losses in chamber experiments (probed in greater detail in
Section 3.4). Biogenic SOA or bV-SOA mass concentrations exceeded 3.2 µg m$^{-3}$ in regions with high biogenic
emissions but were slightly less than 1 µg m$^{-3}$ in urban regions where the POA mass concentrations were the highest.
Previous work has suggested that the bV-SOA in urban regions is formed outside but later transported to the urban
region (Hayes et al., 2015; Heo et al., 2015). Overall, the averaged results over the urban areas appeared to be split
evenly between POA, anthropogenic SOA (aV-SOA+aS-POA+aI-SOA), and biogenic SOA (bV-SOA).
**3.3 Precursor Contributions to OA and SOA**
We examined the absolute OA mass concentrations and precursor contributions to SOA in central Los Angeles across
four different simulations to better understand the effect of successive updates: semi-volatile and reactive POA,
IVOCs, and accounting for vapor wall losses. We chose central Los Angeles as our study area as it is representative of



an urban location with a large population density and suffers from some of the poorest air quality in the United States
(ALA, 2017); results from the sensitivity simulations in Section 3.5 are also discussed at this specific site. Results at
other urban locations (e.g., Riverside, Simi Valley) had similar SOA precursor fractional contributions although the
absolute concentrations did vary a little (see Figure S3). In Figure 3, we plot the 14-day averaged, precursor-resolved
OA mass concentrations and precursor contributions to SOA in Los Angeles from two pairs of four different
simulations. The two pairs represent model predictions based on the low and high $NO_X$ parameterizations.
*Semi-volatile and Reactive POA.* Differences in the Traditional and SVOC simulations were used to highlight the
influence of including a semi-volatile and reactive treatment of POA. The semi-volatile POA treatment resulted in
evaporation of the primary POA emissions from combustion sources (on- and non-road gasoline and diesel,
woodsmoke, biomass burning, and food cooking) and reduced POA mass concentrations by 35% in central Los
Angeles. A ratio of the POA mass concentrations from the SVOC simulation to those from the Traditional simulation
suggested that the POA mass was reduced by approximately 30 to 50% in the urban environment around the central
Los Angeles site (Figure S4). Overall, the POA reductions appeared to be smaller than those implied by the volatility
distributions of May and coworkers (May et al., 2013a, 2013b, 2013c) and those simulated in other atmospheric
models (Robinson et al., 2007). For gasoline, diesel, and biomass burning, May and coworkers (May et al., 2013a,
2013b, 2013c) proposed a 45 to 80% reduction in POA mass concentrations at ambient OA mass concentrations
between 1 and 10 $\mu g\ m^{-3}$. This difference was mainly because we only modeled certain combustion-related POA to be
semi-volatile (i.e., gasoline, diesel, biomass burning, and food cooking sources) while earlier modeling work has
considered POA from all sources to be semi-volatile (e.g., marine, dust). The use of a less volatile food cooking POA
than that used in this work (informed by the works of Woody et al. (2016) and Louvaris et al. (2017)) would tend to
further increase the discrepancy between our work and the findings of May and coworkers. Hu et al. (2014) found that
the combustion sources considered to be semi-volatile in this work accounted for about half of $PM_{2.5}$ mass
concentrations in Los Angeles. While conservative, our POA mass reductions are more realistic until there is evidence
that sources other than those considered here (e.g., marine, dust) produce POA emissions that are semi-volatile too.



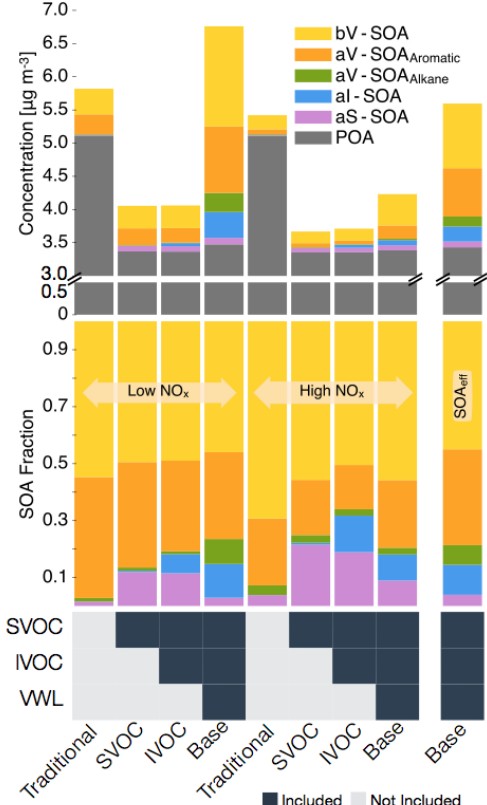

*Figure 3: 14-day averaged model predictions of POA and SOA mass concentrations and precursor contributions at*

*the central Los Angeles site from the sensitivity simulations that examined the influence of updates made in this work.*

*Panel (a) shows absolute concentrations and panel (b) shows precursor contributions. The legend at the bottom*

*tracks how the different pathways (i.e., SOA formation from SVOCs, SOA formation from IVOCs, and correction for*

*chamber vapor wall losses (VWL)) were turned on for the different simulations. Model predictions from the low and*

*high $NO_X$ simulations are shown separately. Model predictions to the extreme right are from accounting for the*

*influence of $NO_X$ on SOA formation using equation 2.*

Allowing the POA vapors or SVOCs to react resulted in only a small fraction of their oxidation products to condense

back as aS-SOA. For example, of the 1.75 μg m$^{-3}$ of POA lost at the central Los Angeles site, only 0.082 μg m$^{-3}$ for

the low $NO_X$ and 0.068 μg m$^{-3}$ for the high $NO_X$ simulations was regained as aS-SOA from oxidation reactions. This

implied a very low chemical conversion efficiency (~4%) for the POA-to-SVOC-to-aS-SOA pump within the urban

area (Miracolo et al., 2010). The SVOCs, at an ambient concentration of 9 μg m$^{-3}$, from gasoline exhaust, diesel

exhaust, and biomass burning emissions had an average carbon number between 18 and 20. Calculations with a box

model version of the SOM suggested that the SOA yields for $C_{18}$ and $C_{20}$ alkanes were between 33 and 86% where the

range includes yields for low $NO_X$ and high $NO_X$ parameterizations. One possible explanation for the difference

between the chemical conversion efficiency in the 3D model and box model yields was that only a small fraction of

the SVOCs had the opportunity to react with OH and form SOA before they were transported out of the urban area.

This does not appear unrealistic. For example, for an SOA precursor with an OH reaction rate constant of $2\times10^{-11}$ cm$^{-3}$

molecules$^{-1}$ s$^{-1}$ and an SOA mass yield of 20%, the chemical conversion efficiency would be ~7% with a daily-





averaged OH concentration of $1.5\times10^6$ molecules $cm^{-3}$ and a reaction time of 4 hours; 4 hours would correspond to a transport of 20 miles at an average wind speed of 5 miles per hour. The central Los Angeles site is only slightly more than 10 miles from the coast.

The low and high $NO_X$ parameterizations had little effect on the aS-SOA mass concentrations presumably because the *n*-dodecane based parameterization used for semi-volatile POA exhibited marginal differences in SOA production under low and high $NO_X$ environments (Loza et al., 2014). Finally, SOA parameterizations based on including the vapor wall loss effect only marginally increased the aS-SOA mass concentrations, especially when viewed in light of the SOA increases from other precursors. We examine the precursor-resolved vapor wall loss effect in more detail in Section 3.4. For the Base simulations, the aS-SOA mass concentrations were a factor of 10 and 2 lower than the aV-SOA mass concentrations for the low and high $NO_X$ parameterizations respectively.

*IVOC.* Differences in the SVOC and IVOC simulations were used to determine the influence of including SOA formation from IVOCs. For both the low and high $NO_X$ simulations, IVOCs contributed marginally to the aI-SOA mass concentrations in Los Angeles (~0.045- μg $m^{-3}$) and elsewhere too (see Figures S3 and S4). The aI-SOA mass concentrations were about half of the aS-SOA mass concentrations for both the low and high $NO_X$ simulations. When compared to the aV-SOA mass concentrations, the aI-SOA mass concentrations were slightly lower for the high $NO_X$ simulations (~40%) but about a factor of five lower for the low $NO_X$ simulations. The inclusion of vapor wall losses seemed to make aI-SOA as or more important than aS-SOA but still less important than aV-SOA; the aI-SOA mass concentrations were a factor of 3.3 and 2.9 lower than the aV-SOA mass concentrations for the Base simulations for the low and high $NO_X$ simulations respectively. Our simulations imply that IVOCs as a bulk class of SOA precursors may not contribute substantially to ambient SOA levels. While they were as influential in forming SOA as SVOCs they were still less important than the traditional SOA precursors (that included long alkanes and aromatics). In this work, the IVOC contribution to SOA was smaller compared to that from traditional SOA precursors mostly because IVOC emissions were only about a third of the traditional SOA precursors (see Section 3.1 for details on emissions). So although IVOCs have higher SOA yields than most of the traditional SOA precursors, the significantly lower IVOC emissions more than offset the increased SOA formation from higher yields. While there are exceptions (e.g., Tsimpidi et al. (2010); Jathar et al. (2017a)), our results did not align with previous box (e.g., Dzepina et al. (2009); Hayes et al. (2015); Ma et al. (2017)) and 3D (e.g., Bergstrom et al. (2012); Zhang et al. (2013)) modeling literature that has found IVOCs to be similar or more important than traditional SOA precursors in contributing to ambient SOA levels. Below we discuss three main reasons for this inconsistency.

First, some previous estimates of IVOC emissions are very likely to be unrepresentative of the in-use gasoline- and diesel-powered sources and unconstrained for biomass burning sources. IVOC emissions in most atmospheric models have previously been determined by scaling emissions of POA or by calculating partitioning with the measured POA, with scaling factors typically on the order of 1.5 (e.g., Shrivastava et al. (2008)) but as large as 3 (e.g., Dzepina et al. (2009)). These factors have been calculated from emissions data from unrepresentative sources: two medium-duty gasoline vehicles built more than two decades ago and a POA volatility distribution from a small off-road diesel engine (Robinson et al., 2007). Additionally, since POA is semi-volatile the POA mass in the particle phase will change with OA loading, which can complicate the use of a scaling based on POA (but this is addressed by the




partitioning method used in some studies). Zhao et al. (2015) provided some evidence for this where they found that the POA-based scaling did not work that well for modern diesel vehicles and instead recommended the use of an NMOG-based scaling. We note that Ma et al. (2017) used the IVOC estimates of Zhao et al. (2015) and still found IVOCs to be comparable to VOCs in terms of SOA production in the Los Angeles area. Second, the SOA formation from IVOCs in most models to date has not been experimentally constrained. Most schemes to model SOA formation from IVOCs have relied on an *ad hoc* aging scheme where IVOCs and their oxidation products react with the OH radical to form lower volatility products with ultimate SOA yields of 100% (Robinson et al., 2007). These schemes do not account for fragmentation reactions and have not been comprehensively validated against experimental data. Jathar et al. (2016) showed that such schemes may significantly overestimate the net aerosol production from SOA precursors. And finally, most models do not use SOA parameters that yet account for the effect of vapor wall losses in chamber experiments. This effect and its particular influence on the IVOC contribution to SOA is discussed in Section 3.4. In this work, we (i) rely on a comprehensive set of IVOC emissions estimates made from measurements performed on representative sources, (ii) model fragmentation reactions during IVOC oxidation, (iii) to some degree constrain SOA formation from IVOCs to chamber experiments, and (iv) to some degree account for the influence of vapor wall losses in chamber experiments. Hence, we argue that our findings on the IVOC contribution to SOA might be more robust than those modeled in earlier studies.

*Traditional VOCs.* For the Base simulations in Los Angeles, aromatics accounted for 33% of the total SOA in Los Angeles and were the most important anthropogenic precursor of SOA. Alkane contributions to SOA were less than 10% for both the low and high $NO_x$ simulations. Biogenic VOCs accounted for 46% and 55% of the total SOA for the low and high $NO_X$ simulations respectively and were clearly the most important precursor of SOA at the central Los Angeles site. After accounting for the influence of $NO_X$ based on equation (2), the isoprene, monoterpene, and sesquiterpene contributions to bV-SOA were 23%, 68%, and 9% respectively, suggesting a strong monoterpene contribution to SOA in southern California. As biogenic VOCs react very quickly with OH and $O_3$ (chemical lifetimes of a few hours), most of the biogenic SOA at this site was likely formed outside the urban airshed and transported to this location, as suggested by Kleeman et al. (2007), Hayes et al. (2015) and Heo et al. (2015).

## 3.4 Influence of Vapor Wall Losses

SOA parameterizations that accounted for the influence of vapor wall losses in chambers seemed to have had a large effect on the absolute mass concentrations of SOA. This can be seen by comparing model results between the IVOC and Base simulations in Figure 3. The SOA mass concentrations were enhanced by a factor of 10.1 and 2.6 for the low and high $NO_X$ simulations respectively and consistent with previous 3D simulations (Cappa et al., 2016). However, they were slightly higher than the range of enhancements reported by Zhang et al. (2014) and estimated by Krechmer et al. (2016) based on analyses of chamber data. The SOA enhancements resulted in an OA enhancement of 1.66 and 1.14 in the low and high $NO_x$ simulations, which were lower than the SOA enhancements since SOA only accounted for a fraction of the OA mass. Differences in enhancements in the low and high $NO_X$ simulations suggest that the vapor wall loss effect was modified by the $NO_X$ level where the enhancement may be lower in urban/source regions with higher $NO_X$ but higher in rural/remote continental regions with lower $NO_X$. Since urban SOA mass concentrations are usually higher than those in rural/remote continental regions, an implication of this $NO_x$-modified enhancement is that accounting for vapor wall loss artifacts will tend to reduce gradients in SOA mass concentrations




between urban and rural/remote continental regions and make SOA more of a regional pollutant similar to ozone ($O_3$).

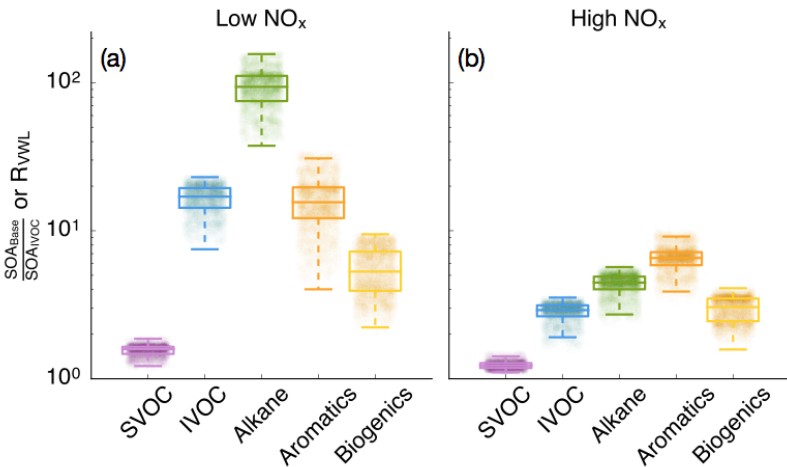

*Figure 4: Ratio of model predictions from the Base simulation that accounts for the influence of vapor wall losses to*
*model predictions from the IVOC simulation that does not account for the influence of vapor wall losses. Ratios are*
*calculated from the 14-day averaged results for the whole domain and are resolved by precursor. Panels (a) and (b)*
*show results from the low and high $NO_X$ simulations respectively.*

Different precursors contributed in varying degrees to the SOA enhancement. The precursor-resolved enhancements
are visualized in Figure 4 where we plot the ratio of the 14-day averaged model predictions of the SOA mass
concentrations from the Base simulation to those from the IVOC simulation for each grid cell in the southern
California domain (dots) and overlay box-whisker plots based on those data. For all precursors the enhancements were
higher for the low $NO_X$ simulations compared to the high $NO_X$ simulations. SVOCs showed the smallest enhancement
at both the low and high $NO_X$ levels (median of 1.6 and 1.2) and hence their fractional contribution to total SOA was
reduced in the Base simulation when compared to the IVOC simulation. Alkanes showed the largest enhancement in
the low $NO_X$ simulations (median of 94) and the second largest enhancement in the high $NO_X$ simulations (median of
4.5). Despite the large enhancements, alkanes still contributed marginally to total SOA in the Base simulations
because the baseline contribution of alkanes to SOA was small in the IVOC simulations (<3%). IVOCs exhibited a
larger enhancement (median of 17 and 2.9) compared to SVOCs and a smaller enhancement compared to alkanes in
both simulations, despite using the same surrogate (i.e., *n*-dodecane) to model SOA formation. The reason for varying
enhancements in SVOC, IVOCs, and alkanes, despite using the same surrogate (i.e., *n*-dodecane), was that the vapor
wall loss-related enhancement was inversely related to the carbon number where larger carbon number precursors
(e.g., SVOC that had an average carbon number of 18 to 20) showed smaller enhancements and smaller carbon
number precursors (e.g., alkanes that included species between carbon numbers of 6 and 12) showed larger
enhancements. The simplest explanation for this inverse relationship is that larger precursors and their oxidation
products, relatively speaking, have shorter chemical lifetimes and undergo fewer chemical reactions before
condensing, which make them less susceptible to being lost to the walls (see Figure S5 where we plot the vapor wall

0    loss-related enhancement in SOA yields as a function of the carbon number at an OA mass concentration of 9 µg m$^{-3}$).

1    Of the two other important precursors, aromatics displayed the largest enhancement in the high $NO_X$ simulations



(median of 6.6) and were tied with IVOCs for the second largest enhancement in the low $NO_X$ simulations (median of 16) while biogenic VOCs showed the lowest enhancement after SVOC in both the low $NO_X$ and high $NO_X$ simulations. Accounting for vapor wall loss artifacts is expected to result in an increase in the aromatic contribution to SOA when compared against biogenic VOCs.

Recent work has argued that vapor wall loss rates in Teflon chambers are much higher (larger than a factor of 5) than by Cappa et al. (2016) and in this work to derive the SOM parameterizations (Huang et al., 2018; Krechmer et al., 2016; Sunol et al., 2018). The use of a higher wall loss rate will tend to increase SOA mass concentrations even further. This new understanding will need to be considered in the future.

## 3.5 Sensitivity Analysis

Results from the sensitivity simulations that examined uncertainties in select model inputs are shown in Figure 5 where we plot the 14-day averaged model predictions from these simulations at the central Los Angeles site. We also plot model predictions from the Base simulations as all the sensitivity simulations have been performed using the Base simulation as the reference (see Table 3 for details about the simulations). Model predictions from the low and high $NO_X$ simulations are shown separately. The No Aging simulations decreased the SOA mass concentrations by almost an order of magnitude demonstrating the importance of modeling multi-generational aging in the SOM. The inclusion of oligomerization reactions that may enhance the partitioning of semi-volatile species may alter this finding. The No-Aging simulations produced a very different precursor contribution to total SOA compared to the Base simulations and the changes in the precursor contribution were also different between the low and high $NO_X$ simulations. For instance, the aV-SOA contributions to total SOA increased from 39% to 41% for the low $NO_X$ simulations but decreased from 26% to less than 5% in the high $NO_X$ simulations. This implied that the treatment of multi-generational aging in the SOM did not proportionately enhance the SOA mass concentrations from the different precursors but rather produced varying levels of enhancement for the different precursors that was further modified by the $NO_X$ levels. This finding is of note because CTMs that have employed schemes such as the volatility basis set (VBS) have typically assumed that multi-generational aging has an approximately similar effect on SOA mass concentrations from different precursors, regardless of the $NO_X$ levels, and one which does not significantly change the precursor contribution to SOA. With the VBS, one may observe some differences with multi-generational aging from the use of different starting VBS distributions for SOA from different precursors.

The $SVOC_{max}$ simulations that assumed all POA (except marine POA) to be semi-volatile saw POA mass concentrations decrease by 36% compared to the Base simulations and by 56% compared to the Traditional simulations (not shown here but inferred from results in Figure 3). The increase in SVOCs from the additional evaporation of POA mass resulted in about a three-fold increase in the aS-SOA mass concentrations and a proportionate increase in the SVOC contribution to total SOA. Similar to the findings discussed in Section 3.3, only a fraction of the POA mass lost was regained as aS-SOA mass concentrations. For instance, when compared to the Traditional simulations, of the 2.9/3.3 µg m$^{-3}$ of POA mass lost 0.32/0.22 µg m$^{-3}$ was regained as aS-SOA reflecting a chemical conversion efficiency of 11/7% for the low/high $NO_X$ simulations. These simulations predicted the maximum decrease in POA mass concentrations from treating all POA as semi-volatile and reactive but still predicted POA to be 40% and 69% of the total OA in the low and high $NO_X$ simulations respectively. This indicated that direct





emissions of POA were still a sizeable fraction of the ambient OA and PM burden.
Estimating IVOCs to be 20% of the NMOG emissions for all combustion sources and modeling the SOA formation
from IVOCs using a $C_{15}$ linear alkane – as modeled in the $IVOC_{max}$ simulations – resulted in an approximately four-
fold increase in the aI-SOA mass concentrations over the Base simulations. The increases were partly attributed to
additional IVOC emissions from sources other than mobile and biomass burning (factor of 2.8 compared to IVOC
emissions from the Base simulations) and partly to using a larger alkane ($C_{15}$ linear alkane) with a higher SOA mass
yield to model SOA formation from IVOCs emitted by gasoline sources. Simulating SOA formation from IVOCs
using an aromatic surrogate in the S-$IVOC_{aromatic}$ simulations had the same effect as the $IVOC_{max}$ simulations and
increased aI-SOA mass concentrations by a factor of 2.6/6.3 for the low/high $NO_X$ simulations. The aI-SOA mass
concentrations were higher because aromatics for the same carbon number have a higher SOA mass yield than
alkanes. The $IVOC_{max}$ and S-$IVOC_{aromatic}$ simulations potentially present an upper bound contribution of IVOCs to
SOA formation and in both these simulations were ~30% of the total SOA and a factor of ~1.5-2 larger than the
aromatic VOC contribution. While the $IVOC_{max}$ and S-$IVOC_{aromatic}$ simulations dramatically increased the aI-SOA
mass concentrations, these simulations only modestly increased the total OA mass concentrations over the low and
high $NO_X$ simulations (average increase of 10%). Over the urban area, the OA mass concentrations in the $IVOC_{max}$
and S-$IVOC_{aromatic}$ simulations were on average 10-12% higher compared to the Base simulations (see Figure S6).
Finally, updating the emissions profiles based on the work of May et al. (2014) had a negligible effect on the SOA
mass concentrations and its precursor contribution implying that the emissions profiles from more than a decade and a
half ago may be sufficient to model the modern mobile source fleet.



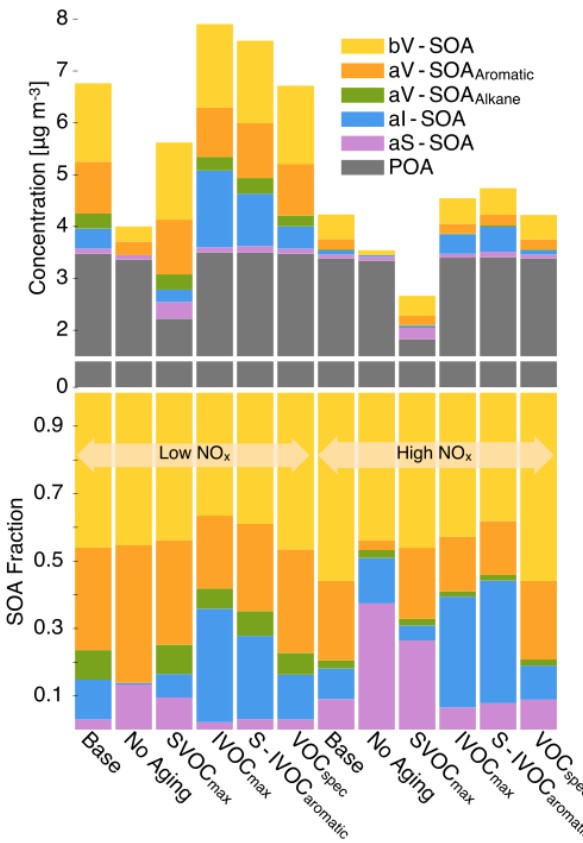

*Figure 5: 14-day averaged model predictions of POA and SOA mass concentrations and precursor contributions from*
*the sensitivity simulations. Panel (a) shows absolute concentrations and panel (b) shows precursor contributions.*
*Model predictions from the low and high $NO_X$ simulations are shown separately. Simulation legend: Base = Base*
*case, No Aging = only models first generation chemistry in the SOM, $SVOC_{max}$ = all POA treated as semi-volatile,*
*$IVOC_{max}$ = all combustion sources assumed to have 20% IVOC emissions and a $C_{15}$ SOA yield, $S\text{-}IVOC_{aromatic}$ =*
*SVOCs and IVOCs modeled as high-yield aromatic compounds, $VOC_{spec}$ = mobile source emissions profiles based on*
*May et al. (2014). More details about these simulation inputs can be found in Section 2.3.*
**3.6 $NO_X$-Adjusted SOA Formation**
The SOM currently does not model the continuous evolution of SOA under varying $NO_X$ concentrations. One of the
challenges in modeling the $NO_X$ influence on SOA formation has been in quantifying the branching of the VOC
oxidation under low and high $NO_X$ conditions. Most commonly used schemes in atmospheric models use the $NO:HO_2$
ratio to determine the initial branching of the precursor to form SOA via the low or the high $NO_X$ pathway. However,
this scheme depends on an accurate prediction of NO and $HO_2$. To assess, at least qualitatively, the ability of the
model to capture NO and $HO_2$ concentrations, we compare 14-day averaged diurnal profiles from this work to those
measured in Pasadena in 2010 during the CalNex campaign in Figure S7. We found that the model predictions were
within a factor of two for NO concentrations but were about a factor of 10 lower than the measured $HO_2^*$
concentrations. We should note that the $HO_2^*$ measurements included $HO_2$ and a fraction of $RO_2$ radicals, where $RO_2$

0  radicals contributed to ~30% of the $HO_2^*$ measurements (Griffith et al., 2016). The inclusion of $RO_2$ should not



change the findings reported here. If the results from our modeling are representative of results from other atmospheric models that use SAPRC or other gas-phase chemical mechanisms, underestimating the $HO_2$ concentrations may lead $NO:HO_2$ ratio-based schemes to overestimate the SOA formed via the high $NO_X$ pathway. Given this limitation and the fact that the SOM does not model the model the continuous evolution of SOA under varying $NO_X$ concentrations, we attempted to model the $NO_X$-dependent SOA formation using $VOC:NO_X$ ratios and $NO_X$ concentrations.

Four different methods – described in equations (1) through (4) – were used to adjust the SOA mass concentrations from each individual precursor to account for the influence of $NO_X$. The adjusted SOA mass concentrations, referred to as $SOA_{eff}$, were summed to calculate the total SOA mass concentrations. Equation (2) produced the highest SOA mass concentrations while equation (3) produced the lowest SOA mass concentrations amongst the four equations. Scatter plots comparing the SOA mass concentrations calculated using equation (2) to those calculated using other equations, in Figure S8, show that the SOA mass concentrations based on equation (2) were, on average, a factor of 1.27, 3.19, and 1.92 higher than those with equations (1), (3), and (4) respectively. This meant that a calculation based on the $VOC:NO_X$ ratio produced a stronger response of $NO_X$ on SOA mass concentrations than the $NO_X$ concentrations themselves. In the subsequent sections, where we evaluate the model predictions (Section 4) and predicted future changes in the OA burden (Section 5), we used the $SOA_{eff}$ calculations based on equation 2 since they represented an upper bound estimate of the $NO_X$ effect on SOA mass concentrations.

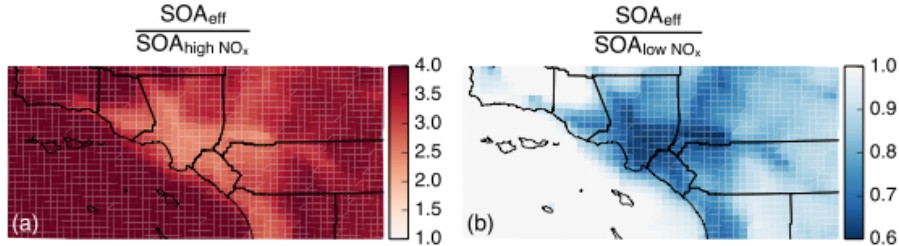

*Figure 6: 14-day averaged ratio of the $SOA_{eff}$ mass concentration to the SOA mass concentration from the (a) high $NO_X$ and (b) low $NO_X$ Base simulations.*

In Figure 6, we plot the ratio of the total $SOA_{eff}$ mass concentrations based on equation (2) to the total SOA mass concentrations from the (a) high $NO_X$ and (b) low $NO_X$ Base simulations. The $SOA_{eff}$ mass concentrations were higher than the SOA mass concentrations predicted using the high $NO_X$ parameterizations, with an average factor of two increase in urban areas and a maximum factor of four increase in non-urban areas. This was because the model predicted $VOC:NO_X$ ratios in the urban areas were higher than the $VOC:NO_X$ ratios produced in the high $NO_X$ chamber experiments and based on equation (2) the SOA mass concentrations were adjusted upwards to include the SOA predicted using the low $NO_X$ parameterizations. The adjustments increased the SOA mass concentrations because the SOA mass concentrations from each precursor were universally higher with the use of the low $NO_X$ parameterizations compared to the high $NO_X$ parameterizations. The $SOA_{eff}$ mass concentrations were 30-40% lower than the SOA mass concentrations predicted using the low $NO_X$ parameterizations in urban areas, suggesting that the $SOA_{eff}$ mass concentrations were approximately midway between the SOA predictions using the high and low $NO_X$ parameterizations. In contrast, the $SOA_{eff}$ mass concentrations were only marginally lower (10-20%) in the non-urban





areas implying that the VOC:NO$_X$ ratios in these regions were very similar to the VOC:NO$_X$ ratios produced in the
low NO$_X$ chamber experiments. In summary, a modest fraction of the SOA mass may be formed through the 'low-
NO$_X$' pathway in high NO$_X$ urban areas, which may result in substantial increases in the predicted SOA mass
concentration when compared against predictions purely based on the use of high NO$_X$ parameterizations. This low-
NO$_X$ SOA will continue to increase in the future as NO$_X$ concentrations are reduced in urban areas through controls on
mobile sources. In contrast, only a small fraction of the SOA mass may be formed through the 'high-NO$_X$' pathway in
low NO$_X$ non-urban areas and the use of a low NO$_X$ parameterization in these regions will only marginally bias model
predictions of SOA mass concentrations.

## 4 Model Evaluation

Model predictions from the Base simulation were evaluated against gas-phase measurements of SOA precursors and
particle-phase measurements of OA mass concentrations and composition. For the particle-phase measurements, we
focused the model evaluation on predictions adjusted for the NO$_X$ influence on SOA formation using equation 2
(logarithmic dependence on VOC:NO$_X$ ratio).
### 4.1 SOA Precursors
In Figure 7(a), we compare 14-day averaged model predictions of aromatic concentrations for our 2005 episode
against measured temporal trends in summer-averaged single-ring aromatic concentrations at three different sites in
Southern California (Los Angeles-North Main Street, Riverside-Rubidoux, and Long Beach) (SCAQMD, 2017);
model predictions of aromatic concentrations are a sum of the benzene, ARO1, and ARO2 concentrations. On the
same figure, we also plot model predictions of aromatic concentrations at Pasadena for our 2005 episode and
measured single-ring aromatic concentrations made at the Pasadena ground site in 2010 as part of the CalNex
campaign (Zhao et al., 2014). The summertime single-ring aromatic concentrations in southern California have
decreased by a factor of 2 to 3 between 2000 and 2011 presumably from regulations that have targeted emissions from
mobile sources. These reductions agreed well with reported temporal trends in carbon monoxide, nitrogen oxides, and
non-methane organic compounds for Los Angeles over the same time period (Warneke et al. (2012); MacDonald et al.
(2013)). Aromatic measurements at Pasadena in 2010 compared well with the 2010 measurements made ~12 km
southwest of Pasadena at the Los Angeles-North Main Street location suggesting that the summer/campaign-averaged
aromatic concentrations were spatially homogeneous over urban Los Angeles and findings from the model-
measurement comparison at a particular site could be generalized for the larger modeled domain. The model-
measurement comparison for aromatics in 2005 was mixed. Concentrations were overpredicted by a factor of ~1.5 at
the Los Angeles-North Main Street and Long Beach sites but agreed well with measurements at Riverside-Rubidoux.
The predictions might have been overestimated because we were using an older emissions inventory developed for the
year 2000 but adapted for use for the year 2005 based on activity data (Hu et al., 2015). Another possibility for the
over prediction was that the lumped model species ARO1 and ARO2 in SAPRC-11 also included emissions from
oxygenated aromatic (e.g., phenols) and aromatic-like compounds (e.g., furans) while the measurements were limited
to a handful of single-ring reduced aromatic compounds. Despite differences in the absolute concentrations, the model
seemed to capture the measured spatial differences between the three sites, i.e. Los Angeles-North Main Street >
Riverside-Rubidoux > Long Beach.




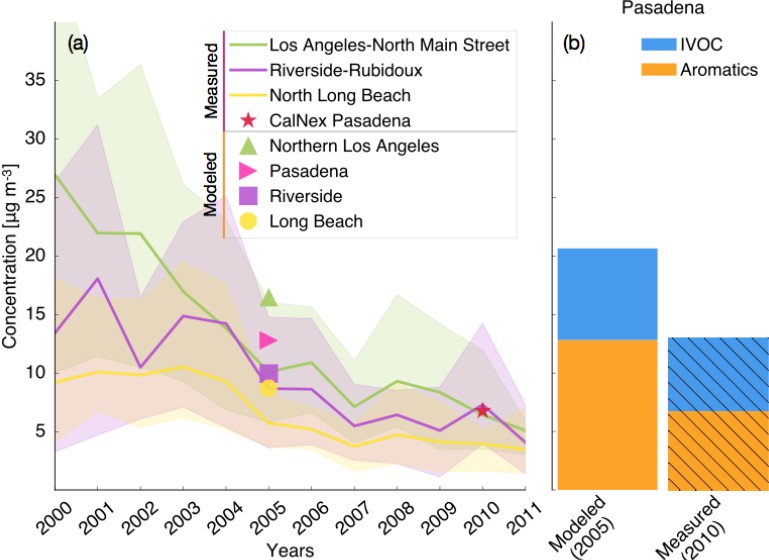

*Figure 7: (a) Mass concentrations of single-ring aromatics in southern California at different sites between 2000 and*

*2011. Measurements show the temporal trend in the summertime mean (solid line) and $10^{th}$-$90^{th}$ percentile (bands) at*

*Los Angeles, Riverside, and Long Beach from 2000 to 2011 (ARB, 2017) and the campaign-averaged measurement*

*from CalNex at the Pasadena ground site in 2010 (Zhao et al., 2014). Model predictions show the 14-day averaged*

*concentration simulated in this work at four different sites (solid symbols) in 2005. (b) Mass concentrations of single-*

*ring aromatics and IVOCs compared between the model predictions from 2005 (this work) to measurements in 2010*

*(Zhao et al., 2014).*

In Figure 7(b), model predictions of aromatics and IVOCs in Pasadena in 2005 are compared against measurements

made at the Pasadena ground site in 2010. The model predictions in Pasadena were calculated by averaging

predictions from the grid cell that contained the Pasadena ground site and the grid cell immediately to the south. This

was done because the ground site location was very close to the cell boundary to the south and the grid cell containing

the Pasadena ground site included mountains to the north of Pasadena that tended to dilute the concentrations in that

grid cell. The measurements in Figure 7(b) included primary IVOCs but did not include the oxygenated IVOCs

measured by Zhao et al. (2014) since the primary IVOCs, according to the authors, relate most closely to IVOC

emissions from mobile sources. The IVOCs included in this work were mostly (>95%) from mobile sources (see

Figure 1) and the hence the comparison with primary IVOCs was appropriate. The model predicted aromatic

concentrations at Pasadena in 2005 were twice the measured aromatic concentrations at Pasadena in 2010. This

2005(modeled)-to-2010(measured) ratio was slightly higher but still consistent with the measured 2005-to-2010 ratio

in aromatic concentrations at the Los Angeles-North Main Street site (1.67). That the 2005(modeled)-to-

2010(measured) ratio for IVOCs in Pasadena was ~1.0 is some evidence that the model predictions of IVOCs might

be underpredicted in 2005, assuming that the ambient IVOC-to-aromatic ratio did not change between 2005 and 2010.

The IVOC$_{max}$ sensitivity simulation (the only sensitivity simulation that modeled an increase in IVOC emissions)

predicted a 2005(modeled)-to-2010(measured) ratio of 3.15 for IVOCs in Pasadena, which was closer to the measured

aromatic concentrations ratios between 2005 and 2010 at the Los Angeles-North Main Street site. This provides



additional evidence for higher IVOC emissions to be included in the model and it is possible that these additional
IVOC emissions might come from volatile chemical products such as pesticides, coatings, cleaning agents, and
personal care products (McDonald et al., 2018). While this model-measurement comparison validates the aromatic
SOA precursors and to some extent the mobile source IVOC SOA precursors, our model does not account for the
oxygenated IVOCs that Zhao et al. (2014) measured and we recommend that future work investigate the sources,
composition, and the SOA potential for these IVOCs.

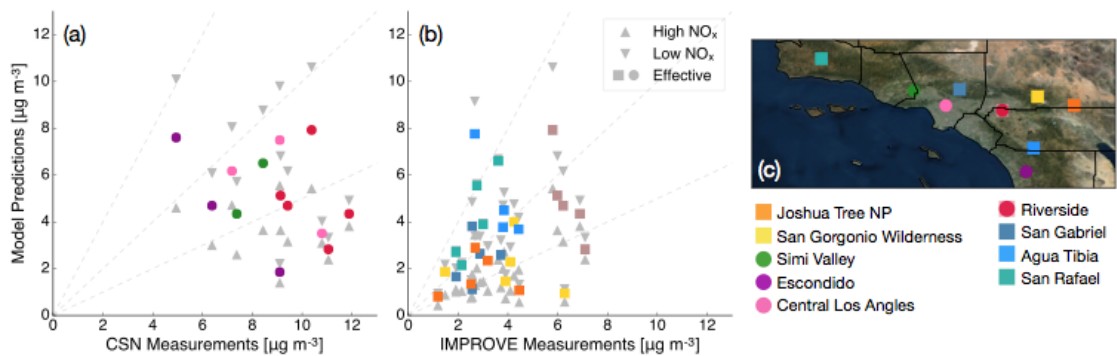

*Figure 8: Model-measurement comparison for daily-averaged OA mass concentrations at (a) CSN and (b) IMPROVE*
*sites in southern California. Panel (c) shows the geographic locations where the comparisons were made.*
**4.2 OA Mass Concentrations**
Scatter plots comparing model predictions of OA from the Base simulations to (a) CSN and (b) IMPROVE
measurements in southern California are shown in Figure 8(a) and (b). Predictions from the low and high $NO_X$
simulations are presented in grey while predictions accounting for the influence of $NO_X$ are shown in color. The
colors denote different sites and the site locations are shown in Figure 8(c). The model-measurement performance is
also captured using statistical metrics of fractional bias, fractional error, and the coefficient of determination in Table
4. At all CSN sites, model predictions of OA that included SOA mass concentrations adjusted for the influence of
$NO_X$ were in-between those predicted between the low and high $NO_X$ simulations. As explained earlier, this was
because the VOC:$NO_X$ ratios at all these sites (see Figure S9(a)) were always higher than those in the high $NO_X$
chamber experiments (see Table 2) and hence the SOA mass concentrations calculated using equation 2 were always
higher than those predicted in the high $NO_X$ simulations. At all the CSN sites, correcting for $NO_X$ improved model
performance compared to the high $NO_X$ experiments but was still inferior compared to the predictions from the low
$NO_X$ simulations (see Table 4). The mean predicted OA mass concentration across all the CSN sites was about 30%
lower than the measurements (5.96 versus 8.86 $\mu g\ m^{-3}$). Model predictions of OA were very similar to those predicted
in the low $NO_X$ simulations at the IMPROVE sites where the VOC:$NO_X$ ratios were higher (e.g., San Rafael–green
square). But, similar to the finding at the CSN sites, model predictions of OA were in-between the predictions
between the low and high $NO_X$ simulations at the IMPROVE sites where the VOC:$NO_X$ ratios were lower as a result
of their proximity to urban areas (e.g., Agua Tibia–blue square and Riverside–brown square). Accounting for $NO_X$
seemed to improve the model performance at the IMPROVE sites when compared to predictions from the high $NO_X$
simulations and were slightly inferior to those from the low $NO_X$ simulations (see Table 4). Of the 27 measurements
available for comparison, 22 or ~80% of the model predictions corrected for $NO_X$ were within a factor of two of





measurements with little bias (fractional bias=-16.63%). Given the differences in the model-measurement comparison
between the CSN (or urban) and IMPROVE (rural/remote continental) sites, the underprediction at the CSN sites
might be indicative of a missing urban source or pathway of OA formation. It is also possible that the urban versus
rural/remote continental difference is an artifact of how the SOM models the oxidation chemistry and/or accounts for
the influence of vapor wall losses. Recently, McDonald et al. (2018) found that volatile chemical products such as
pesticides, coatings, cleaning agents, and personal care products may contribute substantially to IVOC emissions and
account for more than half of the anthropogenic SOA formation in southern California. Our underprediction at urban
sites might be evidence of missing SOA from volatile chemical product-related IVOC emissions. Within the CSN and
IMPROVE sites, we did not find the model-measurement comparison to vary systematically by location. The model-
measurement comparison over all of California using the 24 km simulations produced a similar result (Figure S10).
*Table 4: Statistical metrics of averages, fractional bias, fractional error, and $R^2$ for the model-measurement*
*comparison in southern California.*

| Simulation | CSN | | | | | IMPROVE | | | | |
|---|---|---|---|---|---|---|---|---|---|---|
| | Measured Average ($\mu g\ m^{-3}$) | Modeled Average ($\mu g\ m^{-3}$) | Fractional Bias | Fractional Error | $R^2$ | Measured Average ($\mu g\ m^{-3}$) | Modeled Average ($\mu g\ m^{-3}$) | Fractional Bias | Fractional Error | $R^2$ |
| Base - Low NO$_X$ | 8.86 | 7.96 | -31.5% | 46.0% | 0.16 | 3.72 | 4.87 | -1.38 % | 41.8% | 0.116 |
| Base - Effective | 8.86 | 5.96 | -53.4% | 49.2% | 0.13 | 3.72 | 4.02 | -16.6 % | 44.8% | 0.079 |
| Base - High NO$_X$ | 8.86 | 3.97 | -83.1% | 83.1% | 0.013 | 3.72 | 2.00 | -74.1 % | 75.9% | 0.317 |

Model predictions of the OA:ΔCO diurnal profile and daytime OA versus CO (between 10 am and 8 pm local time)
are compared against measurements made at the Riverside site during the SOAR-1 campaign in Figure 9(a) and (b);
SOA mass concentrations have been adjusted for the influence of NO$_X$ using equation (2). The ΔCO for the
measurements was calculated by assuming a background concentration of 105 ppbv (Hayes et al., 2013) while the
ΔCO for the model predictions was calculated by using the model predicted background concentration of CO over the
ocean to the west of Los Angeles. This model-measurement comparison was not completely coincident in time since
the model results were between July 20 and August 2 while the SOAR-1 campaign spanned from July 15 to August
15. The measurements did not point to any substantial differences in results between the coincident and non-
coincident time and hence we did not anticipate any issues in our comparisons here. The model predictions were able
to capture the general trends in the measured diurnal profile in Figure 9(a) with low ratios during the night, high ratios
attributed to photochemistry in the mid-afternoon, and a peak between 1 and 2 pm (local time). However, the modeled
OA:ΔCO ratios at all times in the diurnal profile in Figure 9(a) and the slope of the OA:CO ratios in Figure 9(b) was
approximately a factor of 2 to 3 lower than the measured ratios, indicating a significant underprediction of urban
SOA, which was consistent with the much higher POA/SOA ratios predicted by the model compared to the
observations, as discussed above. This underprediction cannot be blamed on the model grid resolution since a ratio
with CO should to first order account for the influence of dilution in the grid cell. Cappa et al. (2016) showed much
better model performance than this work when they assumed a non-volatile POA and SOA formed under low NO$_X$
conditions. In this work, despite forming additional SOA from SVOCs and IVOCs, the evaporation of the POA mass
and an SOA estimate adjusted for NO$_X$ meant that the model performance was worse in comparison to Cappa et al.



(2016). The sensitivity simulations of IVOC$_{max}$ and S-IVOC$_{aromatic}$ produced slightly higher OA mass concentrations
(~10-15%) compared to the Base simulations but not dramatically different to influence the comparison in Figure 9(a)
and (b). As mentioned earlier, SOA formation from IVOC emissions from volatile chemical products, or other future
improvements in the SOM, have the potential to reduce the model underprediction at Riverside during the SOAR-1
campaign.

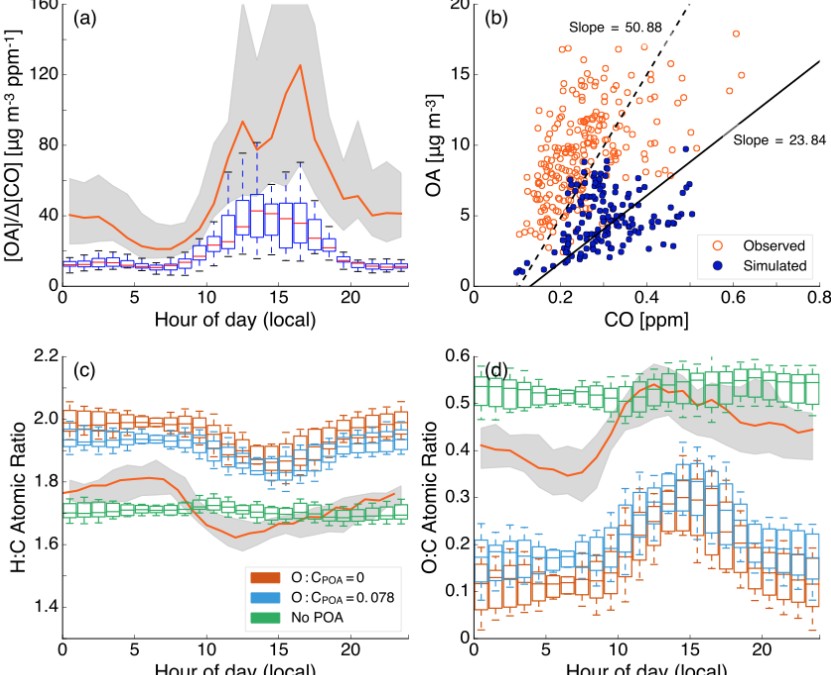

*Figure 9: (a) Diurnal profile of the modeled and measured OA/ΔCO ratios at Riverside, CA. The box plots capture the*
*10th-25th-50th-75th-90th in model predictions over the simulated episode while the gray bands and solid orange line*
*represent the 10th and 90th percentile and median of the measured data. (b) Modeled and measured OA mass*
*concentrations plotted against CO concentrations between 10 am and 8 pm local time. The solid and dashed black*
*lines represent lines fitted to the modeled and measured data by forcing the X-intercept to be the corresponding*
*modeled and measured background CO concentration. Diurnal profiles of the modeled and measured (c) H:C and (d)*
*O:C ratios of the OA (corrected as per Canagaratna et al. (2015)). The three different predictions show results from*
*the Base simulations for OA assuming no change, the POA O:C was fixed to 0.078, and no POA.*
**4.3 POA and SOA Mass Concentrations**
The 14-day averaged results predicted POA and SOA mass concentrations of 3.4 and 2.2 µg m$^{-3}$ and an approximate
60:40 POA-SOA split at Riverside. Docherty et al. (2011) estimated average POA and SOA mass concentrations of
1.9 and 7.0 µg m$^{-3}$ and a POA-SOA split of 20:80 at Riverside during the SOAR-1 campaign. On an absolute basis
model predictions of POA mass concentrations were overpredicted by ~80%. A sensitivity simulation that turned sea
spray emissions off suggested that the 14-day averaged marine POA mass concentrations at Riverside were ~0.8 µg
m$^{-3}$, which are very likely to be overestimated (Hayes et al., 2013). If the emissions of marine POA were updated to





align better with the observations and in the limiting case where the marine POA mass concentrations at Riverside
were negligible, model predicted POA mass concentrations at Riverside (3.4-0.8=2.6 µg m$^{-3}$) would compare well
with the measured values (2.2 µg m$^{-3}$). It is also possible that the model might be over predicting POA because we
only considered POA from certain sources (gasoline and diesel use, woodsmoke, and food cooking) to be semi-
volatile. Figure 1 shows that more than half of the partitioned POA (that excludes marine POA) in southern California
belonged to other sources (e.g., road and construction dust) and this POA was treated as non-volatile in the Base
simulations. Model predictions from the SVOC$_{max}$ simulations that treated all POA except marine POA as semi-
volatile predicted a 14-day averaged POA mass concentration of 2.1 µg m$^{-3}$, which was much closer to the measured
value of 1.9 µg m$^{-3}$. This suggests that all POA, regardless of source, might be semi-volatile and could be modeled so
in atmospheric models. We also compared the hydrocarbon-like OA (HOA) estimate from the measurements, which
was more representative of POA from mobile sources, against model predictions of POA from mobile sources. We did
not model POA from mobile sources separately but if we assumed that mobile sources only accounted for about a
quarter of the partitioned POA mass in southern California (based on Figure 1), our estimated Base model predictions
of POA mass concentrations from mobile sources of 0.85 µg m$^{-3}$ (=3.4×0.25) would be ~30% lower than the measured
HOA mass concentrations of 1.20 µg m$^{-3}$.
On an absolute basis, SOA mass concentrations were underpredicted by a factor of 3 compared to measurements.
Based on the discussion in the previous paragraph, if we added the non-mobile source POA to SOA, the net SOA
mass concentration (3.4×0.75+2.2=4.75 µg m$^{-3}$) was still 33% lower than the measured value. The SOA mass
concentrations in the IVOC$_{max}$ simulations – sensitivity simulations that modeled a fixed IVOC:NMOG ratio of 20%
for all sources except biogenic sources, assumed IVOCs formed SOA similar to a C$_{15}$ linear alkane, and which
produced the maximum SOA mass concentrations amongst all the simulations – were 33% higher than those in the
Base simulation but still ~60% lower than the measured SOA mass concentration of 7 µg m$^{-3}$. A combination of the
two, i.e., adding the non-mobile source POA to the SOA formation in the IVOC$_{max}$ simulations, resulted in a net SOA
mass concentration that was only 22% lower than the measured SOA value. Since the IVOC$_{max}$ simulations produced
ambient IVOC concentrations that were more in line with the measurement trends (see Section 4.1), it is likely that the
IVOC$_{max}$ simulations were better in predicting IVOC concentrations and their contribution to SOA. However, there
are no bottom up (i.e., source) or top down (i.e., atmospheric) data to directly constrain the emissions of and SOA
formation from IVOCs in the IVOC$_{max}$ simulations and hence this finding provides motivation for more detailed
studies of IVOCs in the future.
**4.4 OA Elemental Composition**
The SOM tracks the carbon and oxygen numbers for the OA species and hence we were able to compare model
predictions of the diurnal profiles for the OA H:C and O:C ratios to measurements made at the Riverside site during
the SOAR-1 campaign. The comparisons are shown in Figure 9(c) and (d). For the Base simulations, model
predictions of H:C were significantly overpredicted and those for O:C were significantly underpredicted although the
predictions did capture dips in the H:C and the peaks in the O:C ratios in the mid-afternoon, coincident with peak
photochemical activity. The model predictions did not capture the slight increase in H:C and the decrease in O:C in
the early morning attributed to emissions from rush hour traffic. The high H:C and low O:C predictions were a result
of OA being dominated by POA (~60%), which in this work was modeled as a hydrocarbon distribution that had an



H:C slightly larger than 2.0 and an O:C of 0. Docherty et al. (2011) found that POA had a campaign-averaged H:C of 1.92 and an O:C of 0.078. If the POA O:C were fixed to the values estimated by Docherty et al. (2011), model predictions improved – as shown in Figure 9(c) and (d) – but still under and over predicted the H:C and O:C, respectively; since SOM only tracks carbon and oxygen numbers for an organic species and determines the hydrogen number based on the remaining valence, specifying the O:C dictates the H:C. To assess the ability of the model to predict the elemental composition of SOA, we plot the diurnal profile of H:C and O:C of the SOA in Figure 9(c) and (d). Model predictions of SOA H:C and O:C compared well with the measured range of values but did not reproduce the diurnal changes. Docherty et al. (2011) argued that the H:C and O:C of OA at Riverside was mostly controlled by the SOA composition, which did not change dramatically during the day, and was modified by POA at certain times when POA emissions dominated over SOA production (e.g., nights, rush-hour traffic). This suggests that if absolute predictions of the SOA mass concentrations and the POA-SOA splits were improved, our model would be able to predict both the magnitude and diurnal changes in OA H:C and O:C ratios. We found that the SOA H:C and O:C ratio predictions did not vary significantly and produced similarly flat diurnal profiles across a subset of sensitivity simulations performed (Figure S11), suggesting that the modeled elemental composition of SOA was not very sensitive to the distribution of precursor contributions to SOA.

## 5 Summary and Discussion

Organic aerosol (OA) is an important contributor to urban fine particle pollution yet remains one of its most uncertain components. In this work, we updated the organic aerosol treatment in the UCD/CIT chemical transport model to include a semi-volatile and reactive treatment of POA, emissions and SOA formation from IVOCs, the $NO_X$ influence on SOA formation, and SOA parameterizations for SVOCs and IVOCs that were corrected for vapor wall loss artifacts during chamber experiments. All updates were implemented in the statistical oxidation model (SOM), which simulates the multigenerational aging and gas/particle partitioning of organic aerosol and is embedded in the UCD/CIT model (Cappa et al., 2016; Jathar et al., 2015, 2016). POA, SVOC, and IVOC updates were based on an interpretation of a comprehensive set of source measurements. The influence of $NO_X$ on SOA formation was estimated offline using methods based on the $VOC:NO_X$ ratios/$NO_X$ concentrations.

Despite treating the POA from gasoline, diesel, biomass burning, and food cooking sources as semi-volatile, the updated model only predicted a 30-50% decrease in POA mass concentrations in the urban airshed even when the volatility data used to simulate POA projected a much larger decrease (45 to 80%). The primary reason for the weaker response was that a large fraction of the POA mass came from sources other than those modeled as semi-volatile, e.g., road and construction dust, marine. When all POA, except for marine POA, was modeled as semi-volatile, more than 60% of the POA mass evaporated and the POA mass concentrations under this scenario compared well with measurements made in Riverside, CA as part of the SOAR-1 field campaign. This suggested that all POA, except marine POA, may be modeled as semi-volatile in atmospheric models although we cannot say so with certainty given the limitations of this work in modeling the volatility of food cooking OA and the overestimation of marine POA . Sea spray emissions accounted for a quarter of the POA mass concentrations in the urban airshed but more recent observations suggest that the sea spray emissions or the organic fraction attributed to the sea spray emissions might be overestimated (Hayes et al., 2013). This needs to be examined in future applications of the UCD/CIT model. Atmospheric oxidation of the evaporated POA vapors or SVOCs did not contribute significantly to the SOA burden



($<0.1$ µg m$^{-3}$), even after accounting for the influence of vapor wall loss artifacts, since the timescales for SOA
production appeared to be shorter than the timescales for transport out of the urban airshed.
We found IVOCs to be more important than SVOCs but less important than traditional VOCs such as single-ring
aromatics and biogenics in forming SOA. IVOCs accounted for less than 0.5 µg m$^{-3}$ of SOA while single-ring
aromatics and biogenics each contributed to approximately 1 µg m$^{-3}$ in the Base simulations. The IVOC contribution
to SOA was smaller than that for aromatics partly because IVOC SOA was relatively less sensitive to corrections of
vapor wall loss artifacts in chamber experiments. Another reason for the small IVOC contribution to SOA was that we
only considered IVOC emissions from gasoline, diesel, and biomass burning. On analyzing trends in SOA precursor
concentrations in southern California, the modeled IVOC concentrations in this scenario appeared to be
underpredicted by a factor of ~2. Allowing all sources that emit non-methane organic gases (NMOG) to emit IVOCs
(using an IVOC:NMOG ratio of 0.2) and form SOA similar to a $C_{15}$ linear alkane seemed to increase the IVOC
contribution to SOA (⅓ of total SOA) and produced better comparisons against ambient measurements of IVOC
concentrations, OA composition, and SOA mass concentrations. This might be indicative of missing IVOC emissions
in the model. These missing emissions might be from volatile chemical products such as pesticides, coatings, cleaning
agents, and personal care products, which have been found to contribute substantially to urban SOA burdens
(McDonald et al., 2018). It is also likely that the missing IVOC emissions are from sources considered in this work
(i.e., gasoline, diesel, and biomass burning sources) but were not accounted in the emissions inventories because they
have been shown to be very easily lost to sampling tubes (Pagonis et al., 2017).
Correcting for vapor wall loss artifacts in chamber experiments seemed to increase SOA mass concentrations for all
precursors but the enhancement varied by precursor. With a few exceptions, the SOA enhancements correlated with
carbon number where larger carbon number precursors had lower enhancements and vice versa. The reason for this
inverse relationship was that larger precursors and their oxidation products have shorter chemical lifetimes and
undergo fewer chemical reactions to form SOA, which made them less susceptible to being lost to the chamber walls.
Furthermore, the total SOA enhancement was modified by the NO$_X$ level where low NO$_X$ regions might see higher
enhancements compared to high NO$_X$ regions. In southern California where urban SOA mass concentrations might be
higher than rural/remote continental SOA mass concentrations, the NO$_X$-mediated enhancement will tend to reduce
the spatial gradients in SOA mass concentrations and make SOA a regional pollutant like O$_3$. Accounting for the
influence of NO$_X$ seemed to improve OA model performance against routine measurements in rural/remote
environments (i.e., Interagency Monitoring of Protected Visual Environments network) where OA model predictions
were within a factor of 2 with very little bias (e.g., fractional bias of -16.6%). However, model predictions of OA at
routine monitoring sites in urban environments (i.e., Chemical Speciation Network) and at the Riverside site during
the SOAR-1 field campaign were still underpredicted by at least a factor of 2 (e.g., fractional bias of -49.2%). This
suggested a missing emissions or chemical source of OA in urban areas.



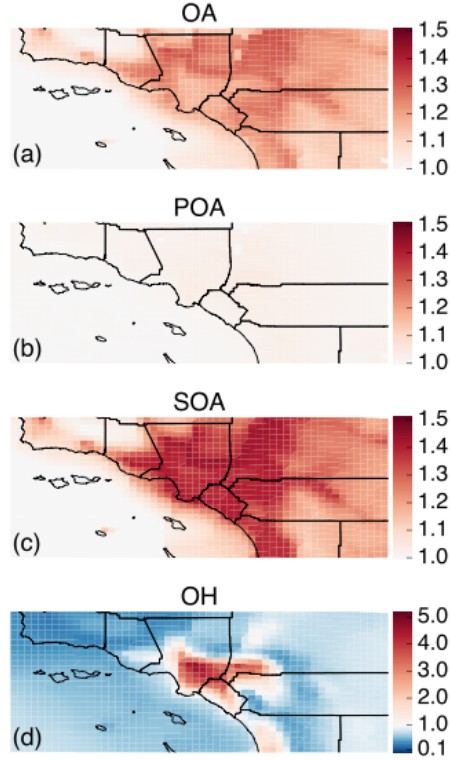

*Figure 10: Ratios of 14-day averaged model predictions of (a) OA, (b) POA, (c) SOA, and (d) OH from 2035 to those*
*from 2005. The 2035 simulations were performed with 2005 meteorological inputs but scaling the anthropogenic*
*emissions for CO, $NO_X$, VOC, $PM_{2.5}$, $SO_2$, and $NH_3$ based on changes projected by the California Emission*
*Projections and Analysis Model (CARB, 2018).*
The future OA burden in southern California will depend not only on reductions in POA and SOA precursor
emissions but also on changes in oxidant concentrations and $VOC:NO_X$ ratios. We used the Base model to simulate
the same time period, July 20 to August 2, for the year 2035 to determine how emissions reductions and atmospheric
conditions may change in a future year to influence ambient OA-POA-SOA mass concentrations. The same
meteorology and environmental conditions were assumed. Emissions reductions in CO, $NO_X$, VOC, $PM_{2.5}$, $SO_2$, and
$NH_3$ were informed by net reductions in statewide emissions between 2005 and 2035 as projected by the California
Emission Projections and Analysis Model (CARB, 2018). The 2005 inventory was scaled based on these emissions
reductions for anthropogenic sources but the biogenic emissions and VOC emissions profiles were kept the same. We
did not resolve the emissions reductions in these pollutants by source or by region since the goal was to examine the
general trend in the OA system and not to predict future air quality; heterogeneity in the reduction in pollutant
emissions by source and geography may alter the results. Statewide emissions reductions in CO, $NO_X$, and VOC of
78%, 83%, and 33% resulted in approximately 50%, 75%, 75%, and 30% reductions in ambient concentrations of CO,
NO, $NO_2$, and VOC in the urban airshed (Figure S12 plots the ratio of CO, NO, $NO_2$, and VOC concentrations in
2035 to those in 2005). Here, VOC is the sum of all organic species tracked in the SAPRC-11 gas-phase chemical
mechanism (excludes methane). Since the $NO_X$ reduction was much more dramatic than that for VOCs, the $VOC:NO_X$
ratio in the urban airshed increased from ~1 to ~5 between 2005 and 2035, which was in line with recent modeled




estimates by Fujita et al. (2016).
We plot the ratio of the mass concentrations for OA, POA, and SOA in 2035 to those in 2005 in Figure 10(a), (b), and
(c) respectively. SOA mass concentrations have been adjusted for the influence of $NO_X$ using equation 2. POA mass
concentrations in the urban airshed in 2035 were slightly higher (~5%) than those in 2005 primarily because $PM_{2.5}$
emissions were higher in 2035 compared to 2005; according to CEPAM, increases in $PM_{2.5}$ emissions were mostly
from increases in area source emissions and not mobile source emissions. Surprisingly, SOA mass concentrations in
the urban airshed were 30-40% higher in 2035 compared to 2005 despite a 30% reduction in VOC emissions and
concentrations. Some of the increase in the SOA mass concentrations was from a shifting $VOC:NO_X$ ratio that
produced more SOA via the low-$NO_X$ pathway. However, the primary reason for the SOA increase was that OH
concentrations in the urban area had increased by a factor of 2 to 4 (see Figure 10(d)) and had reacted more of the
SOA precursors. The OH concentrations were presumably higher in 2035 because lower $NO_X$ emissions resulted in a
higher OH lifetime since the $NO_2$+OH reaction is the primary sink for OH in polluted environments (Jacob, 1999),
including the Los Angeles area (Griffith et al., 2016). These findings suggest that the SOA and OA mass
concentrations may not necessarily respond linearly to reductions in VOC and $NO_X$ emissions in the future but rather
will be strongly influenced by the changes in chemical regime. Similarly, Praske et al. (2018) argue that dramatic
reductions in $NO_X$ emissions and concentrations in urban environments may increasingly lead to SOA formation
through autooxidation pathways and alter the rate and quantify of SOA formed. Hence, attention needs to be paid to
appropriately simulate the chemical regime (e.g., oxidant concentrations, $VOC:NO_X$ ratios, autooxidation reactions) if
we are to accurately simulate the SOA burden in urban environments in the future.

## 6 Acknowledgements

We thank Nehzat Motallebi for sharing the VOC data gathered by the California Air Resources Board in southern
California. AA and SHJ were partially supported by National Oceanic and Atmospheric Administration
(NA17OAR4310003). JLJ was supported by the Environmental Protection Agency (EPA) STAR program (83587701-
0). EPA has not reviewed this manuscript and thus no endorsement should be inferred. SMG, SD, PSS, and CDC were
supported by the National Science Foundation (AGS-0612738, AGS-1104880 and AGS-1523500).

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
