# Peer review of "Simulating secondary organic aerosol in a regional air quality model"

_Atmospheric Chemistry and Physics, 2018_

## Referee Comment (RC1) · Anonymous Referee #1 · 30 Jul 2018

This manuscript describes application of the statistical oxidation model (SOM) to predict OA concentrations in a regional chemical transport model. The modeling focuses primarily on the time period for the 2005 SOAR campaign, though there are also some comparisons to the 2010 CalNex campaign. This is the third in a series of papers regarding application of SOM to regional models, and focuses primarily on impacts of I/SVOCs and NOx on predicted OA concentrations.

[Figure]

Overall the manuscript is appropriate for ACP. However, before publication the authors should work to improve the clarity of presentation. As described in my comments below, the manuscript is at times hard to follow.

One note on the manuscript format: I can't use the line numbers. The pdf I can see has line numbers from 0-9 that repeat. Thus in my comments I try to cite the page number and quote the relevant text where possible.

(1) The manuscript is long and at times hard to follow. While the various topics (e.g., POA partitioning, NOx effects, model-measurement comparison) are placed into organized subsections, there is still a lot of information that the reader needs to keep track of throughout the manuscript. There are nine different case studies (Table 3), each at high and low NOx, and a number of SOA pseudo-species (aI-SOA, aV-SOA_aromatic, etc.). Maybe his level of complexity is unavoidable because of the scope of the study. Nonetheless, I found myself having to go back and forth between the Methods and Results sections.

(2) It's not clear to me what the take-home message of this manuscript is. The most striking result, in my opinion, is shown in Figure 3. This figure shows that vapor wall losses are the largest available "knob" for changing SOA predictions. Including vapor wall losses has a bigger impact on SOA predictions than NOx effects or the inclusion of I/SVOC SOA. Maybe this issue is addressed in more detail in Cappa (2016), but it seems like it deserves more attention in this manuscript. The fact that the SOA predictions are strongly dependent on what amounts to an uncertainty in smog chamber data (because the vapor wall loss is calculated rather than directly sampled) is potentially troubling.

(3) Emissions: I would suggest toning down the rhetoric on whether certain gasoline and diesel vehicle emission profiles are representative for use in chemical transport models. On both page 3 and pages 20-21 the authors are critical of using either the Schauer et al emissions profiles or of scaling POA emissions to estimate IVOCs. It is

a fair criticism that the Schauer emissions profiles are dated (though maybe not too out of date for the 2005 modeling period for SOAR), and that there seem to be better IVOC estimates than scaling POA emissions. However, the authors use the May et al emissions profiles, which include gasoline vehicles up through model year 2010 and DPF-equipped diesel vehicles, and offer no comment on how those profiles might also be inappropriate for a 2005 modeling period. At the very least, it seems like the diesel emissions profile in the model should not include the DPF vehicles tested by May et al, unless there is evidence of significant DPF diesel traffic in California prior to the 2007 change in federal emissions limits for diesels.

(4) NOx effects are included in the final model prediction using equations 1-4. These equations are introduced in the Methods section, but the implications of the various corrections are not discussed. Then on page 16 it is noted that a logarithmic function is used. Since NOx effects are a major focus of this paper, the choice of the logarithmic function needs to be discussed in more detail. For instance, why is a logarithmic function used? Is there a physical basis for using this functional form?

(5) Abstract: "IVOCs did not contribute significantly to SOA mass concentrations in the urban areas" A 15% contribution of IVOCs to SOA does not seem insignificant.

(6) Page 6 "Seven SOM grids were used" I think this means that the SOA from the 9 classes were tracked separately. Please clarify.

(7) What are units of delta_LVP in Table 1?

(8) Bottom of page 7 - alkane seems to be mistyped as "alke"

(9) Page 9 - IVOCs are modeled as either a C13 or C15 alkane, but above and in Table 1 it is stated that IVOCs are modeled as a C12 hydrocarbon. Please clarify.

(10) VOC speciation from May et al - is this only for vehicles relevant to the 2005 fleet?

(11) Figure 2 - It would help to label locations of LA and Riverside.

(12) Page 16 - "In central Los Angeles" - how many grid cells are covered by "central" LA and the whole of LA?

(13) Page 17 compares SOA concentrations in LA and Riverside, and states that a difference of 0.2 ug/m3 of SOA between Riverside and LA is evidence of higher SOA in downwind areas. What is the resolving power of the model (if this were a measurement I would think of the minimum detection limit or precision)? What is the minimum concentration difference that can be claimed as meaningfully different between two locations?

(14) Figure 3 - panels are not labeled as (a) and (b) as noted in the caption.

(15) Last line of page 18, the POA reductions between the Traditional and SVOC model case are "more realistic." More realistic than what? The expected POA reduction from May et al (which was not a modeling study)? It seems like the appropriate comparison for how realistic the model predicted POA concentration is comes from comparing to something like AMS data, not by the fractional evaporation in nonvolatile versus semi-volatile POA cases.

(16) The top of page 30 suggests that missing VOCs from consumer products might be a reason for low predictions at CSN sites. I don't find this explanation very convincing for several reasons. First, the IMPROVE sites are also under predicted (negative bias for both CSN and IMPROVE), so it's not clear there is a missing source of urban VOCs. Figure 3 shows that SOA predictions are most sensitive to vapor wall losses, which could easily account for SOA under predictions. Since adding 80 tons/day of IVOC emissions (Figure 1) barely impacts SOA formation (in the case without vapor wall loss), the corresponding source associated with personal care products would need to be huge to make up for the missing SOA.

(17) Page 33 "under and over predicted the H:C and O:C" I think this is reversed. O:C is under predicted.

---

## Referee Comment (RC2) · Anonymous Referee #2 · 9 Aug 2018

The manuscript describes the update of a chemical transport model, specifically the UCD/CIT model in which the SOM model has been embedded, to predict OA and SOA in California. The modeling period studied is 14 days during late July and early August 2005. The major advance in this work, compared to previous work publish by same group, is the addition of primary IVOCs and SVOCs to the model including treating POA as semi-volatile, while accounting for vapor wall losses during chamber experiments and molecular fragmentation. The authors also examine how NOx levels

impact SOA predictions in the updated model.

The paper is well written, of interest to ACP readers, and appropriate for publication once the following comments have been addressed. I have listed one general comment immediately below followed by more specific and minor comments. Lastly, there is a problem with the manuscript formatting and the line numbers are not displayed correctly in the pdf document. I have thus used the page number and paragraph to indicate the relevant sections of the text.

Length of manuscript: I found the manuscript very long to read, but well-written and clear. I think the scope of the research mostly justifies the length of the manuscript. The sole exception is the section regarding future OA concentrations in 2035. This section felt very much like it was "tacked-on" and it is not well developed. Given the uncertainties in the model, how accurate is the model when projected to 30 years into the future? Sensitivity studies should be run to identify how the predictions vary with different model assumptions as was done for the 2005 simulations. Furthermore, what is the value of running a simulation for only a two week period in 2035? If one wants to inform future policy decisions, it would be better to run the simulation for a much longer period (e.g. a few months). Ultimately, it seems like this work is best left for another manuscript, and deleting this part would also shorten the length of the current manuscript.

Specific Comments:

Introduction, page 3, paragraph 2: The text here stating that SOA formation schemes have been rarely validated against experimental data is too strong or needs to be nuanced. One can, for example, cite the following modeling studies where P-S/IVOCs are treated and models are compared against measurements.

Fountoukis et al. Atmos. Chem. Phys. 2016, 16, 3727-3741.

Murphy et al. Atmos. Chem. Phys. 2017, 17, 11107-11133.

Zhang et al. Atmos. Chem. Phys. 2015, 15, 13973-13992.

Page 8, second paragraph: what is meant by "carbon-equivalent linear alkane"? Is it reasonable to assume that linear alkanes can be used to estimate OH reaction rate constants of SVOCs, given that branched alkanes represent a large portion of POA mass?

Section 2.2.2, last paragraph: Is it correct to state that the model is consistent with Gordon et al.? From the text, it seems that the yields in the model are different versus the Gordon et al. smog chamber studies, but it is expected that the difference in SOA yields will be compensated for by the fact that emissions are different relative to work of Zhao et al.

Section 2.2.3: It would improve the manuscript if the choice of the equation where there is a logarithmic dependence on the VOC/NOx ratio, among the four equations presented, were better justified based on chemical reasons. Currently, the justification is essentially based on the observation that this equation results in the highest SOA prediction.

Table 3: A clarifying question: were the vapor wall losses corrected for VOCs in all the simulations? Even in the traditional case? This is what is indicated by the table, but in the text below it is simply stated "no correction for chamber vapor wall losses", which would seem to exclude also the application of the correction to the VOCs.

Page 19, last sentence: using a 20% yield as an approximate SOA mass yield seems too low, given that earlier in the same paragraph estimated yields for SVOC oxidation ranged from 33% to 86%. I also think the oxidation rate constant is a little low, given that octadecane and nonadecane (for example) have rate constants that are greater than 2e-11 cm-3 molecules-1 s-1, and these compounds likely represent a lower limit as oxidation rates increase with alkane branching. Also, how was the wind speed of 5 miles per hour chosen?

Page 20: The statement saying IVOCs as a bulk class of SOA precursors may not contribute substantially to ambient SOA levels is too strong. There is an important contribution, especially if one only considers anthropogenic SOA, even though that contribution may be less than that from traditional VOCs.

Page 21, first paragraph, last sentence: I don't disagree with the value of the work presented in the manuscript, but, in my opinion, what is really unique is the incorporation of these 4 elements into a chemical transport model. It seems that should be mentioned somewhere in this sentence.

Page 32, Line 3: I think there is a typo here and the measured value given is incorrect and should be 1.9 rather than 2.2.

Page 32, first paragraph, last sentence: The comparison of HOA to POA from mobile sources is rather good. It would be worthwhile to point that out.

Figure S5: Why do the b-alkanes have an enhancement of less than 1 under high NOx conditions? Doesn't correcting for the wall losses always increase the SOA yield?

Minor comments:

Table 1: Simply for the sake of clarity, the order of the molar yields should be indicated. Do they progress (left to right) from the addition of 1 to 4 oxygen atoms per reaction, or is it the opposite order?

Table 2: There is an "&" symbol in the footnotes of the table, but I cannot find the matching symbol in the table or table caption.

Page 7: There appears to be a typo on the last line of this page.

Section 2.2.3: Were the IVOCs and gas phase SVOCs used to calculate the modeled VOC:NOX ratios? These compounds would contribute to the HO2 budget, although likely less than the VOCs.

Page 16, last paragraph: a reference should be provided for the measured mass concentrations of POA over the open ocean west of California.

Page 28: What was the measured aromatic concentration ratio between 2005 and 2010 at the Los Angeles-North Main Street site?

Page 29, last sentence: it should be clarified that the 27 measurements available for comparison are measurements taken at IMPROVE sites. (At least I think that is the case, it is not entirely clear from the manuscript. )

Page 34, lines 1 – 2: There may be a typo here. I thought the argument was the timescales for SOA formation are LONGER than the timescales for transport out of the urban airshed.

Page 36, references: The reference from the American Lung Association doesn't seem to be correct as it contains a link to a website about air quality in Fort Collins, Colorado. In addition, the organization name is shortened as if it is an author's name.

Supporting information: Some of the tables and figures and their matching captions are split across different pages. For readability, each figure and table should appear entirely on one page with it's caption.

---

## Referee Comment (RC3) · Anonymous Referee #3 · 13 Aug 2018

Reviewer comments on "Simulating secondary organic aerosol in a regional 1 air quality model using the statistical oxidation model – Part 3: Assessing the influence of semi-volatile and intermediate volatility organic compounds and NOX," acp-2018-616

This work describes an update to the UCD/CIT chemical transport model, specifically by the inclusion of the SOM organic aerosol model. The impacts of various improvements are described and investigated, such as the inclusion of SVOC and IVOC oxida-

tion. In general, the methods are well described, and the results are reasonable. The manuscript is well within the scope of the journal, and warrants publication with minor revisions as described below.

Major comments: 1) The manuscript is fairly lengthy and detailed. While this is reasonable for a detailed description of model improvements, it can make it difficult to keep track of the main points at times. I wonder if some of the results could be boiled down to really the key points, and some of the subtleties moved to the supplement.

2) As a chemistry-minded member of our community, a frequent point of confusion for me is in discussion of IVOC with lack of precision around the chemical composition being represented. In this case, IVOC is being used to mean non-speciated intermediate volatility combustion emissions, which are primarily branched and cyclic aliphatics and some aromatics. Yet they are represented by n-alkanes (which likely differ in their SOA yields), and are somehow grouped separately than "long alkanes and aromatics", despite this being a reasonable description of these IVOCs (I presume the latter refers to speciated emissions?). It would be helpful to be more precise in the language and consider in the discussion what these emissions likely are.

Technical comments: This line numbering approach is maddening bordering on useless. Please in the future use unique line numbers for all lines on a given page, or better yet use continuous line numbers throughout the document. When I copy and paste into notepad, I see digits before the final appear and are continuous, so perhaps this is just a conversion issue?

Page 1, paragraph 1: extra space before "gas/particle"

Page 3, line 3: remove comma after "formed"

Page 3, paragraph 1: IVOCs are not necessarily unique from "aromatics and long alkanes", in fact as the authors point out that is in large part what they contain (plus cylic and polycyclic aliphatics), so the wording seems a bit off. I would also direct the

authors to Gentner et al., PNAS, 2012 (doi: 10.1073/pnas.1212272109), for a detailed analysis of the composition of combustion emissions in the IVOC range (e.g. diesel fuels), and that work in general that suggested substantial OA formation from diesel fuel components (i.e. IVOCs). Similarly Worton et al. (ES&T, 2014, doi: 10.1021/es405375) found POA from all sources to "look like" motor oil, which was heavily cyclized and branched.

Page 4, paragraph 2: 500 ppbv is very high NOx indeed - are these general trends applicable to more ambient-relevant conditions?

Page 4, paragraph 3: Should be "i.e., Henze", because the authors mean "in other words", not "for example"

Method descriptions of UCD/CIT and SOM are clear and well-written

Page 6, paragraph 2: Though dodecane probably has an approximately appropriate volatility and chain-length, the true chemical composition of combustion related I/SVOCs contains much for branching and cyclization. For future work I would recommend generating an SOM grid for a branched alkylcyclohexane or some similar such compound, if possible.

Page 7, last line: misspelled alkane

Page 8, paragraph 2: Gentner et al. (2012) estimated that branching and cyclization, which likely dominate I/SVOCs decrease SOA yields by a factor of around 3 (based on compiled chamber data available at that time). The assumption of a linear alkane SOM grid could consequently have a significant impact on SOA produced in this work. The authors point to Gentner and Caravaggio to justify that "alkanes" are the substantial fraction, but this somewhat obscures the fact that these alkanes are not linear, which may be important

Page 14, paragraph 1: I'm a bit confused, if measured OA:OC is 1.8-2.1, but are the authors using 1.6?

Figure 1 and discussion therefore: The terminology of splitting "IVOC" and "long alkane and aromatics" is a bit confusing, since the IVOCs are being modeled as long alkanes, and is comprised of alkanes and aromatics. I would recommended something more like "speciated" and "unspeciated", or "lumped". It's not totally clear to me what is a long alkane, and what is an IVOC, but perhaps I missed it in the methods?

Page 17, end of paragraph 1: What might explain this underestimation in SOA? Particularly given that the use of linear alkanes as proxies likely overestimates the SOA yield of some of these groups? Do the later changes to the model fix this regional underestimation?

Page 17, end of paragraph 1: missing a period

Page 20, paragraph 3: "Our simulations imply that IVOCs as a bulk class of SOA precursors may not contribute substantially to ambient SOA levels." Again, I'm not quite sure what to make of this statement, as long alkanes may include species that would be considered IVOCs, so I'm a bit confused by imprecision in language around these compound classes.

Page 22: The authors discuss the impact of faster reaction times, but what about the impact of high volatility preventing wall loss. Does a C6 compound really suffer substantial wall loss, given its ability to re-partition to the gas phase? I would expect most losses to be centered in the IVOC range (less reversibly absorbing to the walls)

Page 26, paragraph 2: The discussion of equations 1-4 is a bit unclear. It would be helpful to remind the reader of the implications of each equation.

Figure 8, Table 4 and discussion thereof: What does it mean from a practical sense that the correlations here are so poor. While I acknowledge the biases and averages are not unreasonable, the model appears unable to capture the temporal variability of these measurements - is that simply due to poor resolution in emissions databases, or is there additional important complexity being ignored? The concern of course is that it

might be telling us something more fundamental about the assumptions or applications of the model.

Figure 9: It's not clear why an O:C of 0.078 was chosen. I imagine it comes from the measurements, but it's not obvious where this is stated. Update: I see it is stated later, this should be brought forward to discussion of the figure or the caption.

Page 34 paragraph 3: Given how big a role vapor wall loss correction plays in the model, it would be helpful to have some discussion of how exactly it is being corrected for. This manuscript just references previous studies, but it warrants some overview here.
* * *

---

## Author Response (AR1)

We thank all three reviewers for their comments. We have revised the manuscript based on their comments and queries and provided a point-by-point response below. Reviewer comments are in regular black, our response is in blue, text from the manuscript is in red, and additions/updates are in *italic magenta*.

**Reviewer 1**

This manuscript describes application of the statistical oxidation model (SOM) to predict OA concentrations in a regional chemical transport model. The modeling focuses primarily on the time period for the 2005 SOAR campaign, though there are also some comparisons to the 2010 CalNex campaign. This is the third in a series of papers regarding application of SOM to regional models, and focuses primarily on impacts of I/SVOCs and NOx on predicted OA concentrations. Overall the manuscript is appropriate for ACP. However, before publication the authors should work to improve the clarity of presentation. As described in my comments below, the manuscript is at times hard to follow.

1. One note on the manuscript format: I can't use the line numbers. The pdf I can see has line numbers from 0-9 that repeat. Thus in my comments I try to cite the page number and quote the relevant text where possible.

We apologize that the line numbers did not translate well when converting the .docx file to a .pdf file.

2. The manuscript is long and at times hard to follow. While the various topics (e.g., POA partitioning, NOx effects, model-measurement comparison) are placed into organized subsections, there is still a lot of information that the reader needs to keep track of throughout the manuscript. There are nine different case studies (Table 3), each at high and low NOx, and a number of SOA pseudo-species (aI-SOA, aV-SOA_aromatic, etc.). Maybe his level of complexity is unavoidable because of the scope of the study. Nonetheless, I found myself having to go back and forth between the Methods and Results sections.

While we agree with the reviewer that the paper is long, the length and complexity of the paper stems from the detail in describing and providing context to the model predictions and the model evaluation. As the reviewer points out, the length to a certain extent is unavoidable. In the reviewed version of the manuscript, we have already summarized the findings from the model in the summary and discussion section (Section 5). To help the reader navigate the paper, we have added a glossary early in the manuscript for reference and have added the following text at the end of the introduction: "*To help the reader, we provide a brief overview of the different sections in this manuscript. Section 2 discusses details of the chemical transport model (2.1), organic aerosol model (2.2), simulations performed (2.3), and measurements used for model evaluation (2.4). In Section 3, we first describe the emissions (3.1), spatial distribution (3.2), and precursor contributions to OA (3.3), followed by the influence of vapor wall losses (3.4) and NO$_X$ (3.6) on SOA formation. In the same section, we describe results from sensitivity simulations performed on the most sensitive inputs (3.5). Next, we compare model predictions of SOA precursors (4.1), OA (4.2), POA, and SOA (4.3) mass concentrations, and OA elemental composition (4.4) against measurements in southern California. Finally, we highlight key findings from this work in the summary and discussion section (5).*".

3. It's not clear to me what the take-home message of this manuscript is. The most striking result, in my opinion, is shown in Figure 3. This figure shows that vapor wall losses are the largest available "knob" for changing SOA predictions. Including vapor wall losses has a bigger impact on SOA predictions than NOx effects or the inclusion of I/SVOC SOA. Maybe this issue is addressed in more detail in Cappa (2016), but it seems like it deserves more attention in this manuscript. The fact that the SOA predictions are strongly dependent on what amounts to an uncertainty in smog chamber data (because the vapor wall loss is calculated rather than directly sampled) is potentially troubling.

As summarized in Section 5, there are several take-home messages from this manuscript, many of which agree with findings in previous literature: (i) treating the POA as semi-volatile will reduce ambient POA mass concentrations, (ii) parameterizations that have been corrected for vapor wall losses will increase ambient SOA mass concentrations, (iii) S/IVOCs, after accounting for the influence of vapor wall losses, do not contribute as much to the SOA burden as traditional VOC precursors (e.g., aromatics), (iv) accounting for the influence of $NO_X$ may increase SOA mass concentrations in high $NO_X$/urban regions and (v) updates included in this work seem to improve the model-measurement comparison for OA mass and composition in southern California. As the reviewer points out, the SOA mass concentrations were substantially enhanced after accounting for the influence of vapor wall losses and this was previously discussed in our previous publication (Cappa et al., 2016). But the vapor wall loss finding does not diminish the importance of the other findings surrounding S/IVOCs and $NO_X$.

4. Emissions: I would suggest toning down the rhetoric on whether certain gasoline and diesel vehicle emission profiles are representative for use in chemical transport models. On both page 3 and pages 20-21 the authors are critical of using either the Schauer et al emissions profiles or of scaling POA emissions to estimate IVOCs. It is a fair criticism that the Schauer emissions profiles are dated (though maybe not too out of date for the 2005 modeling period for SOAR), and that there seem to be better IVOC estimates than scaling POA emissions. However, the authors use the May et al emissions profiles, which include gasoline vehicles up through model year 2010 and DPF-equipped diesel vehicles, and offer no comment on how those profiles might also be inappropriate for a 2005 modeling period. At the very least, it seems like the diesel emissions profile in the model should not include the DPF vehicles tested by May et al, unless there is evidence of significant DPF diesel traffic in California prior to the 2007 change in federal emissions limits for diesels.

We agree with the reviewer that our criticism with the old data came out too strong. We have updated the text in the introduction (Section 1) and the results (Section 3.3) as follows.
Section 1: "*Models have assumed that these data are representative of emissions from modern diesel-powered sources and the POA/IVOC properties from diesel sources are similar to those from other sources. New source data are now available to update the POA and IVOC emissions estimates in chemical transport models.*".
Section 3.3: Added the qualifier "*likely to be less representative*" instead of "very likely to be unrepresentative" and "*performed on more representative sources*" instead of "performed on representative sources".

The reviewer raises an important point of the compatibility of using the May et al. (2013a, 2013b) data that has sources manufactured after 2005 to inform the POA and IVOC emissions estimates for the vehicle/engine fleet in 2005. For all the data used in this work, there is either no

evidence or very little data to suggest that any of the inputs used in this work were different for sources manufactured before and after 2005. This is now made clear with the following additions in Section 2.2:

- *"Almost three-quarters of the light-duty gasoline vehicles used in May et al. (2013a) were manufactured in or prior to 2005 (the year modeled in this work) and they did not find the POA volatility distribution data to be sensitive to the model year of the vehicle. Hence, the volatility distribution used in this work should still be representative of the vehicle fleet in 2005."*
- *"May et al. (2013b) did not report on differences in the POA volatility distribution between vehicles that did or did not use a modern emissions control system (diesel particulate filter (DPF) and/or diesel oxidation catalyst (DOC)). Hence, we assumed that the volatility distribution used here was still representative of the mostly non-DPF and non-DOC vehicle fleet in 2005."*
- *"The IVOC:NMOG fractions did not appear to be statistically different for the gasoline and diesel sources manufactured before or after 2005 and hence those fractions were assumed to be representative of the source fleet in 2005."*

5. NOx effects are included in the final model prediction using equations 1-4. These equations are introduced in the Methods section, but the implications of the various corrections are not discussed. Then on page 16 it is noted that a logarithmic function is used. Since NOx effects are a major focus of this paper, the choice of the logarithmic function needs to be discussed in more detail. For instance, why is a logarithmic function used? Is there a physical basis for using this functional form?

We have cited previous work to justify our choice for the use of $VOC:NO_X$ and $NO_X$ to model the $NO_X$ dependence on SOA formation and provided explanations for the use of linear and logarithmic formulations. The text in Section 2.2.3 was modified as follows: *"Presto et al. (2006) found that the SOA from α-pinene ozonolysis under varying $NO_X$ conditions could be estimated by interpolating the SOA formed between the low and high $NO_X$ conditions using the $VOC:NO_X$ ratio.* Hence, in the first method, we used the $VOC:NO_X$ ratios from the low and high $NO_X$ chamber experiments as our bounds and used the 3D model predicted $VOC:NO_X$ ratio to interpolate between the minimum and maximum SOA mass concentrations predicted from the low and high $NO_X$ simulations. *Previous work (e.g., Camredon et al. (2007), Xu et al. (2015)) has also found SOA formation to vary along a $NO_X$ scale and hence, i*n the second method, we used $NO_X$ concentrations from the low and high $NO_X$ chamber experiments and the 3D model predictions to perform the interpolation. For each method, we performed the interpolation on the SOA mass concentrations assuming a linear or logarithmic dependence on the $VOC:NO_X$ ratios and $NO_X$ concentrations. The linear dependency was chosen for simplicity while the logarithmic dependency was chosen to mimic the visual trends in SOA and $VOC:NO_X$ or $NO_X$ reported in previous work."*. The choice of equation 2 (i.e., logarithmic dependence of $VOC:NO_X$ on SOA) for Sections 4 and 5 was based on the SOA being most sensitive to that formulation. None of the equations proposed in Section 2.2.3 have been validated and we added the following sentence to Section 3.6 to make that clear: *"The validity of equation 2 needs to be examined in future work."*.

6. Abstract: "IVOCs did not contribute significantly to SOA mass concentrations in the urban areas" A 15% contribution of IVOCs to SOA does not seem insignificant.
We have reworded the text in the abstract to: "Model predictions suggested that both SVOCs (evaporated POA vapors) and IVOCs did not contribute *as much as other anthropogenic*

*precursors (e.g., alkanes, aromatics)* to SOA mass concentrations in the urban areas (<5% and <15% of the total SOA respectively) as the timescales for SOA production appeared to be shorter than the timescales for transport out of the urban airshed.".

7. Page 6 "Seven SOM grids were used" I think this means that the SOA from the 9 classes were tracked separately. Please clarify.
SOA from 9 different precursor types or classes was modeled but tracked in 7 different SOM grids. Monoterpenes and sesquiterpenes were tracked on the same grid since they both used the same SOM parameters to model SOA formation. Similarly, SVOCs and IVOCs were tracked on the same grid. The following text is reproduced from the paper that describes the SOM grid setup: "Seven SOM grids were used to represent SOA formation from nine different precursor classes: (i) long alkanes, (ii) benzene, (iii) high-yield aromatics, (iv) low-yield aromatics, (v) isoprene, (vi) monoterpenes, (vii) sesquiterpenes, (viii) semi-volatile POA (SVOC), and (ix) IVOCs. Classes (i) through (vii) have been included in previous applications of the SOM and we refer the reader to our earlier publications for more details (Cappa et al., 2016; Jathar et al., 2015, 2016). Classes (viii) and (ix) were included in this work for the first time. The SOA formation from monoterpenes and sesquiterpenes (classes vi and vii) was modeled in the same SOM grid since both precursors used the SOM parameter sets for α-pinene. Similarly, the SOA formation from SVOCs and IVOCs was modeled in the same SOM grid and both used the SOM parameter sets for n-dodecane; sensitivity simulations were performed using the SOM parameter set for toluene.".

8. What are units of delta_LVP in Table 1?

$\Delta$LVP has units of [$\log_{10}$] $\mu g/m^3$. It is the change in vapor pressure with the addition of one oxygen to the carbon backbone. To link the SOM parameters in Table 1 to the text, we have updated the text as follows: "Six precursor-specific adjustable parameters are assigned for each SOM grid: four parameters that define the molar yields of the four functionalized, oxidized products ($P_{func}$), one parameter that determines the probability of functionalization or fragmentation ($m_{frag}$) and one parameter that describes the relationship between $N_C$, $N_O$ and volatility ($\Delta LVP$).".

9. Bottom of page 7 - alkane seems to be mistyped as "alke"

We have corrected the 'alke' to '*alkane*'.

10. Page 9 - IVOCs are modeled as either a C13 or C15 alkane, but above and in Table 1 it is stated that IVOCs are modeled as a C12 hydrocarbon. Please clarify.

Following Jathar et al. (2014), IVOCs from the three combustion sources (gasoline exhaust, diesel exhaust, and biomass burning) are modeled as a $C_{13}$ or $C_{15}$ alkane. However, the SOM parameters to model the oxidation chemistry for $C_{13}$ or $C_{15}$ (or any other alkane for that matter) are based on fits from chamber experiments performed on a $C_{12}$ alkane. As the SOM informs the general oxidation scheme and trajectory through the carbon-oxygen grid, the SOA mass yields for $C_{12}$, $C_{13}$, and $C_{15}$ on using the same parameter set for $C_{12}$ could be different. We have reproduced text from the manuscript that explains this point for SVOCs but one that is equally applicable to IVOCs: "The reactive behavior of POA was modeled by assuming that the POA vapors (i.e. SVOCs) (represented as a hydrocarbon distribution) and their products participated

in gas-phase oxidation and formed SOA similar to linear alkanes and utilized the SOM parameter set for n-dodecane. The surrogate, in this case n-dodecane, only informs the multi-generational oxidation chemistry of the precursor and the actual compound of interest (e.g., a $C_{15}$ linear alkane) can have a different SOA mass yield than that of n-dodecane.". For completeness, however, we have added the following text when describing the methods to model SOA formation from IVOCs: "*As mentioned earlier, n-dodecane, only informs the multi-generational oxidation chemistry of the precursor and the actual compound of interest (e.g., a $C_{13}$ or $C_{15}$ linear alkane) can have a different SOA mass yield than that of n-dodecane.*".

11. VOC speciation from May et al - is this only for vehicles relevant to the 2005 fleet?

May et al. (2014) did not find the VOC speciation to differ substantially with the vehicle model year and hence the VOC speciation used in this work was representative of the 2005 vehicle fleet.

12. Figure 2 - It would help to label locations of LA and Riverside.

We have added labels denoting the two sites to Figure 2.

[Figure]

13. Page 16 - "In central Los Angeles" - how many grid cells are covered by "central" LA and the whole of LA?

The central Los Angeles site is not referring to an area/region but rather to an EPA CSN (chemical speciation network) site located in 'central' Los Angeles. We have added the following text in parentheses in a few instances where we refer to the central Los Angeles site "*(grid cell containing the CSN site)*".

14. Page 17 compares SOA concentrations in LA and Riverside, and states that a difference of 0.2 ug/m3 of SOA between Riverside and LA is evidence of higher SOA in downwind areas. What is the resolving power of the model (if this were a measurement I would think of the

minimum detection limit or precision)? What is the minimum concentration difference that can be claimed as meaningfully different between two locations?

The resolving power of the model, purely from a numerical perspective, should be quite high (say less than 0.001 µg m⁻³). The resolving power of the model should be lower as we consider the uncertainty associated with representing the emissions, chemistry, transport, and deposition in the chemical transport model, all of which will tend to affect the spatial distribution of SOA. We have not done any systematic model experiments to study the resolving power of the model so, as the reviewer points out, we do not know if the 0.2 µg m⁻³ difference between the LA and Riverside sites is evidence for higher SOA in downwind areas. However, we should note that the SOA differences between LA and Riverside are retained across our suite of sensitivity simulations. We have altered the text to reflect the reviewer's comment: "SOA mass concentrations, *in contrast* to POA, had a more regional presence *with lesser differences between the upwind and downwind regions* (e.g., 2.4 µg m⁻³ in Riverside versus 2.2 µg m⁻³ in central Los Angeles) or in regions with high emissions of biogenic VOCs (e.g., 2.5 µg m⁻³ inside the Los Padres National Forest).".

15. Figure 3 - panels are not labeled as (a) and (b) as noted in the caption.

[Figure]

This has been corrected.

16. Last line of page 18, the POA reductions between the Traditional and SVOC model case are "more realistic." More realistic than what? The expected POA reduction from May et al (which was not a modeling study)? It seems like the appropriate comparison for how realistic the model

predicted POA concentration is comes from comparing to something like AMS data, not by the fractional evaporation in nonvolatile versus semivolatile POA cases.

*The word 'realistic' was used here to indicate that we have only considered POA to be semi-volatile from sources where there is evidence for such (i.e., gasoline, diesel, biomass burning, and food cooking sources) and kept the POA from other sources as non-volatile. We have rephrased the sentence as follows: "The POA mass reductions shown here are conservative and might have been larger if there was evidence that sources other than those considered here (e.g., marine, dust) produced POA that was semi-volatile too.". The model predicted POA was compared to the AMS measurements in Section 4.3 to assess how realistic the predictions are.*

17. The top of page 30 suggests that missing VOCs from consumer products might be a reason for low predictions at CSN sites. I don't find this explanation very convincing for several reasons. First, the IMPROVE sites are also under predicted (negative bias for both CSN and IMPROVE), so it's not clear there is a missing source of urban VOCs. Figure 3 shows that SOA predictions are most sensitive to vapor wall losses, which could easily account for SOA under predictions. Since adding 80 tons/day of IVOC emissions (Figure 1) barely impacts SOA formation (in the case without vapor wall loss), the corresponding source associated with personal care products would need to be huge to make up for the missing SOA.

*For the 'Base-Effective' model results (and the 'Base-Low $NO_X$' model results), the model-measurement comparison at the IMPROVE sites is better (fractional bias of -16.6%) than at the CSN sites (fractional bias of -53.4%). So, it is possible that the difference in the model-measurement comparison can be attributed to an urban source of SOA. We cite the McDonald et al. (2018) paper because this recent paper makes strong arguments that atmospheric models are missing an important source of SOA in urban areas. We do not think it is the only explanation for the urban-rural differences in the model-measurement comparison and we state in the same paragraph that, "It is also possible that the urban versus rural/remote continental difference is an artifact of how the SOM models the oxidation chemistry and/or accounts for the influence of vapor wall losses.". Furthermore, as the reviewer would agree, there are large uncertainties surrounding the IVOC emissions and their potential to form SOA from volatile chemical products (VCP) so it might be premature to say that the differences observed in this work stem from not including SOA formation from VCP-IVOCs. Accordingly, we have added rearranged the text in that paragraph to give the impression that VCP-IVOCs might be one reason to explaining the urban-rural differences in the model-measurement comparison: "Given the differences in the model-measurement comparison between the CSN (or urban) and IMPROVE (rural/remote continental) sites, the underprediction at the CSN sites might be indicative of a missing urban source or pathway of OA formation. Recently, McDonald et al. (2018) found that volatile chemical products such as pesticides, coatings, cleaning agents, and personal care products may contribute substantially to IVOC emissions and account for more than half of the anthropogenic SOA formation in southern California. Our underprediction at urban sites might be evidence of missing SOA from volatile chemical product-related IVOC emissions. However, it is also possible that the urban versus rural/remote continental difference is an artifact of how the SOM models the oxidation chemistry and/or accounts for the influence of vapor wall losses.".*

18. Page 33 "under and over predicted the H:C and O:C" I think this is reversed. O:C is under predicted.

This has been corrected.

**Reviewer 2**

The manuscript describes the update of a chemical transport model, specifically the UCD/CIT model in which the SOM model has been embedded, to predict OA and SOA in California. The modeling period studied is 14 days during late July and early August 2005. The major advance in this work, compared to previous work publish by same group, is the addition of primary IVOCs and SVOCs to the model including treating POA as semi-volatile, while accounting for vapor wall losses during chamber experiments and molecular fragmentation. The authors also examine how NOx levels impact SOA predictions in the updated model.

The paper is well written, of interest to ACP readers, and appropriate for publication once the following comments have been addressed. I have listed one general comment immediately below followed by more specific and minor comments. Lastly, there is a problem with the manuscript formatting and the line numbers are not displayed correctly in the pdf document. I have thus used the page number and paragraph to indicate the relevant sections of the text. Length of manuscript: I found the manuscript very long to read, but well-written and clear. I think the scope of the research mostly justifies the length of the manuscript.

1. The sole exception is the section regarding future OA concentrations in 2035. This section felt very much like it was "tacked-on" and it is not well developed. Given the uncertainties in the model, how accurate is the model when projected to 30 years into the future? Sensitivity studies should be run to identify how the predictions vary with different model assumptions as was done for the 2005 simulations. Furthermore, what is the value of running a simulation for only a two week period in 2035? If one wants to inform future policy decisions, it would be better to run the simulation for a much longer period (e.g. a few months). Ultimately, it seems like this work is best left for another manuscript, and deleting this part would also shorten the length of the current manuscript.

The primary motivation to perform the future air quality modeling was to raise awareness in the research and regulatory community that SOA formation in the future may be influenced not only by changes in VOC emissions and VOC:$NO_X$ ratios but also by changes in oxidant (e.g., OH) concentrations. Zhao et al. (2017) recently contended that SOA formation in southern California may increase in the future with changes in the VOC:$NO_X$ ratios, despite decreases in SOA precursor emissions. They used a box model to draw those conclusions. Through the use of the air quality model developed in this work, we imply that increases in the OH concentrations from reductions in $NO_X$ could be much more important in determining ambient SOA mass concentrations than changes in the VOC:$NO_X$ ratios. Our future air quality modeling results are not intended to accurately capture the absolute concentrations in a future world but rather to communicate the sensitivity of future SOA to changes in emissions, chemical regimes, and oxidant loadings. Sensitivity simulations similar to those performed for the 2005 episode would not change the percentage changes in OA-POA-SOA mass concentrations. Our model results contribute to this evolving discussion since they include the latest knowledge about SOA formation in a 3D model framework.

2. Introduction, page 3, paragraph 2: The text here stating that SOA formation schemes have been rarely validated against experimental data is too strong or needs to be nuanced. One can, for example, cite the following modeling studies where P-S/IVOCs are treated and models are compared against measurements.
Fountoukis et al. Atmos. Chem. Phys. 2016, 16, 3727-3741.
Murphy et al. Atmos. Chem. Phys. 2017, 17, 11107-11133.
Zhang et al. Atmos. Chem. Phys. 2015, 15, 13973-13992.

We agree with the reviewer. We have revised the text to say that some studies have validated the SOA formation schemes against experimental data and included the recommended citations: "*While some of these schemes have been validated against experimental data (Fountoukis et al., 2016; Hodzic and Jimenez, 2011; Murphy et al., 2017; Zhang et al., 2015), most have assumed that all sources have the same rate and potential to form SOA and, in some cases, ignore fragmentation reactions tied to multigenerational chemistry.*".

3. Page 8, second paragraph: what is meant by "carbon-equivalent linear alkane"?

'Carbon-equivalent' here means that the POA hydrocarbon included in the model has the same reaction rate constant with OH as a linear alkane with the same number of carbon atoms.

4. Is it reasonable to assume that linear alkanes can be used to estimate OH reaction rate constants of SVOCs, given that branched alkanes represent a large portion of POA mass?

The reviewer is right to point out that if the POA mass is mostly comprised of branched/cyclic alkanes (and perhaps even aromatics), as we point out later in that paragraph, the reaction rate constants with OH would be larger than the carbon-equivalent linear alkane. We acknowledge this limitation in the following sentence: "*We should note that the presence of branched/cyclic alkane and aromatic compounds in the SVOCs would require the use of a higher reaction rate constant with OH as these compounds are more reactive with OH than carbon-equivalent linear alkanes.*".

5. Section 2.2.2, last paragraph: Is it correct to state that the model is consistent with Gordon et al.? From the text, it seems that the yields in the model are different versus the Gordon et al. smog chamber studies, but it is expected that the difference in SOA yields will be compensated for by the fact that emissions are different relative to work of Zhao et al.

Based on the work of Jathar et al. (2014), the linear alkane surrogate used to model the IVOCs from gasoline, diesel, and biomass burning sources was chosen to reproduce the SOA formation observed in chamber data (Gordon et al., 2014a, 2014b; Hennigan et al., 2011). This was stated earlier through the following sentence: "The equivalent linear alkane to model SOA formation from IVOCs in Jathar et al. (2014) was based on fitting the SOA formation observed in chamber experiments *(Gordon et al., 2014a; Gordon et al., 2014b; Hennigan et al., 2011)* and hence the choice of the hydrocarbon in this work was experimentally constrained.". We have added the citations for the chamber studies to be clear. The two subsequent paragraphs describe the differences of our IVOC-SOA model with the work of Zhao et al. (2015, 2016).

6. Section 2.2.3: It would improve the manuscript if the choice of the equation where there is a logarithmic dependence on the VOC/NOx ratio, among the four equations presented, were better

justified based on chemical reasons. Currently, the justification is essentially based on the observation that this equation results in the highest SOA prediction.

We have cited previous work to justify our choice for the use of VOC:NO$_X$ and NO$_X$ to model the NO$_X$ dependence on SOA formation and provided explanations for the use of linear and logarithmic formulations. The text in Section 2.2.3 was modified as follows: "*Presto et al. (2006) found that the SOA from α-pinene ozonolysis under varying NO$_X$ conditions could be estimated by interpolating the SOA formed between the low and high NO$_X$ conditions using the VOC:NO$_X$ ratio.* Hence, in the first method, we used the VOC:NOX ratios from the low and high NO$_X$ chamber experiments as our bounds and used the 3D model predicted VOC:NO$_X$ ratio to interpolate between the minimum and maximum SOA mass concentrations predicted from the low and high NO$_X$ simulations. *Previous work (e.g., Camredon et al. (2007), Xu et al. (2015)) has also found SOA formation to vary along a NOX scale and hence, i*n the second method, we used NO$_X$ concentrations from the low and high NO$_X$ chamber experiments and the 3D model predictions to perform the interpolation. For each method, we performed the interpolation on the SOA mass concentrations assuming a linear or logarithmic dependence on the VOC:NO$_X$ ratios and NO$_X$ concentrations. The linear dependency was chosen for simplicity while the logarithmic dependency was chosen to mimic the visual trends in SOA and VOC:NO$_X$ or NO$_X$ reported in previous work and also to produce the highest response in the SOA formation with NO$_X$.". The choice of equation 2 (i.e., logarithmic dependence of VOC:NO$_X$ on SOA) for Sections 4 and 5 was based on the SOA being most sensitive to that formulation. None of the equations proposed in Section 2.2.3 have been validated and we added the following sentence to Section 3.6 to make that clear: "*The validity of equation 2 needs to be examined in future work.*".

7. Table 3: A clarifying question: were the vapor wall losses corrected for VOCs in all the simulations? Even in the traditional case? This is what is indicated by the table, but in the text below it is simply stated "no correction for chamber vapor wall losses", which would seem to exclude also the application of the correction to the VOCs.

The vapor wall loss corrected SOM parameterizations were either applied for all SOA precursors or not at all. This is now clarified in the caption for Table 3.

8. Page 19, last sentence: using a 20% yield as an approximate SOA mass yield seems too low, given that earlier in the same paragraph estimated yields for SVOC oxidation ranged from 33% to 86%. I also think the oxidation rate constant is a little low, given that octadecane and nonadecane (for example) have rate constants that are greater than 2e-11 cm-3 molecules-1 s-1, and these compounds likely represent a lower limit as oxidation rates increase with alkane branching. Also, how was the wind speed of 5 miles per hour chosen?

We thank the reviewer for this comment. The reviewer is correct to point out that we need to use a higher SOA mass yield. The way the calculation was done assumed that all of the SOA from SVOC oxidation in the Los Angeles grid cell was from 20 miles away. As the central Los Angeles site is only 10 miles from the coast and the prevailing winds are from west to east, our calculation represented an upper bound on the chemical conversion efficiency. We have revised our calculation and the text to represent the minimum and maximum chemical conversion efficiencies assuming that the SOA from SVOC oxidation arose from SVOC emissions in the same grid cell to up to 2 grid cells away (up to 12.5 miles away). The wind speed of 5 miles per hour was based on the average wind speed in central Los Angeles in the month of July. In our

revised calculation and updated text, we have updated the wind speed to 5.4 miles per hour based on data gathered from Weather Spark (a website that collates meteorological information). The update text is as follows: "*If we assume that most of the sS-SOA in the grid cell that contains the Los Angeles site was from the oxidation of SVOCs released in that grid cell and from grid cells that are up to two grid cells away, our results do not appear unrealistic. For example, for an SOA precursor with an OH reaction rate constant of $2.4 \times 10\text{-}11\ cm^{-3}$ molecules$^{-1}$ s$^{-1}$ (average value from a $C_{18}$ and $C_{20}$ linear alkane) and an SOA mass yield of 60% (average from the SOA mass yield range described earlier for a $C_{18}$ and $C_{20}$ linear alkane), the chemical conversion efficiency would be 3.5-15% with a daily-averaged OH concentration of $1.5 \times 10^6$ molecules cm$^{-3}$ and a reaction time of 0.5-2.3 hours. A reaction time of 0.5 to 2.3 hours corresponds to a transport of 2.5 (half a grid cell) and 12.5 (2.5 grid cells) miles at an average wind speed of 5.4 miles per hour (Weather Spark).*".

9. Page 20: The statement saying IVOCs as a bulk class of SOA precursors may not contribute substantially to ambient SOA levels is too strong. There is an important contribution, especially if one only considers anthropogenic SOA, even though that contribution may be less than that from traditional VOCs.

The sentence the reviewer mentions and the following sentence have been combined to change the tone as follows: "*Our simulations imply that IVOCs might be as influential as SVOCs as a bulk class of SOA precursors but they were still less important than the traditional SOA precursors (that included long alkanes and aromatics) in contributing to ambient SOA levels.*".

10. Page 21, first paragraph, last sentence: I don't disagree with the value of the work presented in the manuscript, but, in my opinion, what is really unique is the incorporation of these 4 elements into a chemical transport model. It seems that should be mentioned somewhere in this sentence.

We agree with the reviewer on this point and have updated the sentence as follows: "In this work, we (i) rely on a comprehensive set of IVOC emissions estimates made from measurements performed on representative sources, (ii) model fragmentation reactions during IVOC oxidation, (iii) to some degree constrain SOA formation from IVOCs with chamber experiments, (iv) to some degree account for the influence of vapor wall losses in chamber experiments, *and (v) include all of the previously mentioned updates in a chemical transport model.*".

11. Page 32, Line 3: I think there is a typo here and the measured value given is incorrect and should be 1.9 rather than 2.2.

This has been corrected.

12. Page 32, first paragraph, last sentence: The comparison of HOA to POA from mobile sources is rather good. It would be worthwhile to point that out.

The wording was changed to: "We did not model POA from mobile sources separately but if we assumed that mobile sources only accounted for about a quarter of the partitioned POA mass in southern California (based on Figure 1), our estimated Base model predictions of POA mass concentrations from mobile sources of 0.85 µg m$^{-3}$ (=3.4×0.25) *would compare reasonably with the* measured HOA mass concentrations of 1.20 µg m$^{-3}$.".

13. Figure S5: Why do the b-alkanes have an enhancement of less than 1 under high NOx conditions? Doesn't correcting for the wall losses always increase the SOA yield?

The SOM parameters for branched alkanes were based on chamber experiments performed on methylundecane. These parameters, relatively speaking, indicate a higher propensity to fragment since the $P_{frag}$ value is lower and the functionalized products have more oxygens added to the carbon backbone per reaction step. This suggests that in the absence of vapor losses to the chamber wall fragmentation will become progressively more important during a chamber experiment. Hence, the most likely explanation for an enhancement less than 1 for smaller carbon number branched alkanes is that the chamber wall absorbs and protects some of the oxidation products from being fragmented.

Minor comments:

14. Table 1: Simply for the sake of clarity, the order of the molar yields should be indicated. Do they progress (left to right) from the addition of 1 to 4 oxygen atoms per reaction, or is it the opposite order?

Thanks for the comment. We have added the labels $Pf_1$ through $Pf_4$ below the label $P_{func}$ to the make this clear. We have also added this detail in the caption.

15. Table 2: There is an "&" symbol in the footnotes of the table, but I cannot find the matching symbol in the table or table caption.

The '&' applies to the calculation of the VOC:$NO_X$ ratio for all low $NO_X$ experiments. The table reproduced below captures that change.

*Table 2: Low and high VOC:$NO_x$ ratios in ppb ppb$^{-1}$ from chamber experiments used to model the influence of $NO_X$ on SOA formation.*

| SOM surrogate | (VOC:NO$_X$)$_{low}$ NOx | NO$_{X,low}$ NOx | (VOC:NO$_X$)$_{high}$ NOx | NO$_{X,high}$ NOx | Reference |
|---|---|---|---|---|---|
| n-dodecane | 17.0[&] | <2 ppbv | 0.09 | 343 | Loza et al.(2014) |
| benzene | 207[&] | <2 ppbv | 1.98 | 169 | Ng et al.(2007) |
| toluene | 46.3[&*] | <0.8 ppbv | 0.76[*] | 50 | Zhang et al.(2014) |
| m-xylene | 12.1[&#] | <2 ppbv | 0.10 | 943 | Ng et al.(2007) |
| isoprene | 24.5[&] | <2 ppbv | 0.29 | 937 | Chhabra et al.(2010) |
| α-pinene | 33.1[&] | <2 ppbv | 0.05 | 844 | Chhabra et al.(2010) |

[&]*minimum VOC:$NO_x$ ratios since these assume a $NO_X$ concentration of 0.8 ppbv in the chamber*
[*]*average of six experiments performed by Zhang et al.(2014)*
[#]*average of two experiments performed by Ng et al.(2007)*

16. Page 7: There appears to be a typo on the last line of this page.
The typo has been corrected.

17. Section 2.2.3: Were the IVOCs and gas phase SVOCs used to calculate the modeled VOC:NOX ratios? These compounds would contribute to the HO2 budget, although likely less than the VOCs.

Yes, the gas-phase IVOCs and SVOCs were used to calculate the modeled VOC:NOx ratios. We have added this detail to the text: "For the VOC:NO$_{X,model}$ ratio, the VOC is the sum of all organic species tracked in the SAPRC-11 gas-phase chemical mechanism, *including all IVOCs and gas-phase SVOCs*.".

18. Page 16, last paragraph: a reference should be provided for the measured mass concentrations of POA over the open ocean west of California.

The correct reference for this Hayes et al. (2013) and this has been added to the end of that sentence.

19. Page 28: What was the measured aromatic concentration ratio between 2005 and 2010 at the Los Angeles-North Main Street site?

As stated in the parentheses, the measured aromatic concentration ratio between 2005 and 2010 at the Los Angeles-North Main Street site was 1.67. This ratio was consistent with the 2005(modeled)-to-2010(measured) ratio of aromatic concentrations of 2 at Pasadena.

20. Page 29, last sentence: it should be clarified that the 27 measurements available for comparison are measurements taken at IMPROVE sites. (At least I think that is the case, it is not entirely clear from the manuscript.)

This is now made clear that the 27 measurements are at the IMPROVE sites: "Of the 27 *IMPROVE* measurements available for comparison, 22 or ~80% of the model predictions corrected for NOX were within a factor of two of measurements with little bias (fractional bias=-16.63%).".

21. Page 34, lines 1 – 2: There may be a typo here. I thought the argument was the timescales for SOA formation are LONGER than the timescales for transport out of the urban airshed.

Yes, that is correct. The sentence has been corrected.

22. Page 36, references: The reference from the American Lung Association doesn't seem to be correct as it contains a link to a website about air quality in Fort Collins, Colorado. In addition, the organization name is shortened as if it is an author's name.

We apologize for this oversight. This has been corrected.

23. Supporting information: Some of the tables and figures and their matching captions are split across different pages. For readability, each figure and table should appear entirely on one page with it's caption.

This has been fixed.

**Reviewer 3**

Reviewer comments on "Simulating secondary organic aerosol in a regional 1 air quality model using the statistical oxidation model – Part 3: Assessing the influence of semi-volatile and

intermediate volatility organic compounds and NOX," acp-2018-616 This work describes an update to the UCD/CIT chemical transport model, specifically by the inclusion of the SOM organic aerosol model. The impacts of various improvements are described and investigated, such as the inclusion of SVOC and IVOC oxidation. In general, the methods are well described, and the results are reasonable. The manuscript is well within the scope of the journal, and warrants publication with minor revisions as described below.

Major comments:

1. The manuscript is fairly lengthy and detailed. While this is reasonable for a detailed description of model improvements, it can make it difficult to keep track of the main points at times. I wonder if some of the results could be boiled down to really the key points, and some of the subtleties moved to the supplement.

While we agree with the reviewer that the manuscript is lengthy, the length was dictated by the breadth of findings discussed. The key findings are brought together and summarized in the 'Discussion' section, alongside its implications for the atmospheric modeling of OA in urban environments. The 'Discussion' section hence provides an overview of the key results of this study. To help the reader navigate the manuscript, we have added the following text at the end of the introduction: "*To help the reader, we provide a brief overview of the different sections in this manuscript. Section 2 discusses details of the chemical transport model (2.1), organic aerosol model (2.2), simulations performed (2.3), and measurements used for model evaluation (2.4). In Section 3, we first describe the emissions (3.1), spatial distribution (3.2), and precursor contributions to OA (3.3), followed by the influence of vapor wall losses (3.4) and NOX (3.6) on SOA formation. In the same section, we describe results from sensitivity simulations performed on the most sensitive inputs (3.5). Next, we compare model predictions of SOA precursors (4.1), OA (4.2), POA, and SOA (4.3) mass concentrations, and OA elemental composition (4.4) against measurements in southern California. Finally, we highlight key findings from this work in the summary and discussion section (5).*".

2. As a chemistry-minded member of our community, a frequent point of confusion for me is in discussion of IVOC with lack of precision around the chemical composition being represented. In this case, IVOC is being used to mean non-speciated intermediate volatility combustion emissions, which are primarily branched and cyclic aliphatics and some aromatics. Yet they are represented by n-alkanes (which likely differ in their SOA yields), and are somehow grouped separately than "long alkanes and aromatics", despite this being a reasonable description of these IVOCs (I presume the latter refers to speciated emissions?). It would be helpful to be more precise in the language and consider in the discussion what these emissions likely are.

We agree with the reviewer. We have added more detail in Section 2.2.2 to be clear about we define, add, and model IVOCs in this work as a new SOA precursor and how the IVOCs added to the model in this work are distinct from the speciated long alkane and aromatic precursors already present in the emissions inventory. We reproduce part of that section will updates here: "In Jathar et al. (2014), IVOC emissions, defined as the sum of all unspeciated compounds, were determined as a mass fraction of the total non-methane organic gas (NMOG) emissions for three different source categories: gasoline vehicles, diesel vehicles, and biomass burning. *Here, the IVOCs, as unspeciated organic compounds, are new SOA precursors added to the emissions inventory and regardless of their chemical makeup are distinct from the speciated precursors*

*such as long alkanes and aromatics already present in existing emissions inventories. …* The equivalent linear alkane to model SOA formation from IVOCs in Jathar et al. (2014) was based on fitting the SOA formation observed in chamber experiments and hence the choice of the hydrocarbon in this work was experimentally constrained. *Jathar et al. (2014) used linear alkanes as a surrogate as the SOA formation from linear alkanes was well studied when they developed the parameterization and the SOA mass yields increased predictably with the carbon number of the precursor. Recent application of gas-chromatography mass-spectrometry to combustion emissions has found that IVOCs are mostly composed of branched/cyclic alkane and aromatic compounds (Gentner et al., 2012; Koss et al., 2018; Zhao et al., 2016, 2017). So while it would have been more appropriate to model the IVOCs as an alkane-aromatic mixture, this choice would not have substantially changed the model predictions in the work as the SOA formation from this alkane-aromatic mixture would still be constrained to the same chamber experiments. We will consider the recent detailed speciation work surrounding IVOCs in future applications of this model. In this work,* we also investigated the sensitivity in model predictions to the use of an aromatic compound (i.e., toluene) as a surrogate instead of an alkane (i.e., n-dodecane) to model SOA formation from IVOCs (see rationale in Section 2.4).".

The following text was added to Section 5 to motivate future work with detailed speciation now available for IVOCs from some combustion sources: "*The IVOCs in this work were modeled using a linear alkane surrogate despite recent evidence that IVOCs in combustion emissions are a mixture of branched and cyclic alkanes, aromatics, and oxygenated compounds with very few linear alkanes (Koss et al., 2018; Zhao et al., 2016, 2017). A more chemically appropriate representation of the IVOCs would not have substantially changed the findings in this work since the linear alkane surrogates were chosen to reproduce the SOA formation in chamber experiments performed on combustion emissions. However, future work should incorporate the more detailed speciation available to model the emissions and SOA formation from IVOCs.*".

3. This line numbering approach is maddening bordering on useless. Please in the future use unique line numbers for all lines on a given page, or better yet use continuous line numbers throughout the document. When I copy and paste into notepad, I see digits before the final appear and are continuous, so perhaps this is just a conversion issue?

We apologize that the line numbers did not translate well when converting the .docx file to a .pdf file.

4. Page 1, paragraph 1: extra space before "gas/particle"

This has been corrected.

5. Page 3, line 3: remove comma after "formed"

This has been corrected.

6. Page 3, paragraph 1: IVOCs are not necessarily unique from "aromatics and long alkanes", in fact as the authors point out that is in large part what they contain (plus cyclic and polycyclic aliphatics), so the wording seems a bit off. I would also direct the authors to Gentner et al., PNAS, 2012 (doi: 10.1073/pnas.1212272109), for a detailed analysis of the composition of combustion emissions in the IVOC range (e.g. diesel fuels), and that work in general that

suggested substantial OA formation from diesel fuel components (i.e. IVOCs). Similarly Worton et al. (ES&T, 2014, doi: 10.1021/es405375) found POA from all sources to "look like" motor oil, which was heavily cyclized and branched.

The reviewer is right to point out that the sentences were not consistent. They have been corrected as follows and now include a citation to Gentner et al. (2012): "In addition, all combustion processes are now believed to include emissions of an important additional class of SOA precursors: intermediate-volatility organic compounds (IVOCs) (Jathar et al., 2014). Gas-chromatography mass-spectrometry applications have suggested that they are primarily composed of high molecular weight *linear, branched, and cyclic* alkanes (carbon numbers greater than 12) and aromatics (*Gentner et al., 2012*; Zhao et al., 2014, 2017).". Thank you for mentioning the Worton et al. (2014) study. It has already been mentioned later (Section 2.2.2) when explaining the rationale of using alkanes as a surrogate to model the chemistry and gas/particle partitioning of POA/SVOCs.

7. Page 4, paragraph 2: 500 ppbv is very high NOx indeed - are these general trends applicable to more ambient-relevant conditions?

We apologize for the error but the parentheses should have read '>50 ppbv' instead of '>500 ppbv'. Previous 'high $NO_X$' chamber experiments have been performed across a range of initial $NO_X$ values spanning from 50 ppbv to 1 ppmv. We have modified the text in the parentheses to '*>50 ppbv and up to ~1 ppmv*'. To answer the reviewer's question, the trends in SOA mass yields with $NO_X$ seem applicable to even atmospherically-relevant concentrations of $NO_X$ since chamber experiments performed at even modestly high $NO_X$ concentrations (~100 ppbv) produce differences in SOA mass yields between low and high $NO_X$ experiments (e.g., for toluene as shown in Zhang et al. (2014)). What still remains unclear, however, is how the SOA mass yields vary over a continuous range of initial $NO_X$ concentrations and how changes in the $NO_X$ concentrations during the chamber experiment or transport from high $NO_X$ to low $NO_X$ environments affect the SOA mass yield.

8. Page 4, paragraph 3: Should be "i.e., Henze", because the authors mean "in other words", not "for example".

This has been corrected.

9. Page 6, paragraph 2: Though dodecane probably has an approximately appropriate volatility and chain-length, the true chemical composition of combustion related I/SVOCs contains much for branching and cyclization. For future work I would recommend generating an SOM grid for a branched alkylcyclohexane or some similar such compound, if possible.

We agree with the reviewer's comment that S/IVOCs, based on previous emissions characterization work, are more likely to be branched or cyclic alkanes rather than linear alkanes and that parameterizations based on a branched/cyclic alkane would have been more appropriate for use in our work. We accept this to be a limitation of the current work and in future work we will plan to use parameterizations for branched/cyclic alkanes (e.g., SOM parameters developed for methylundecane and hexylcyclohexane by Cappa et al. (2013)) to model the SOA formation from SVOCs. We have added some more detail to Section 2.2.2 and Section 5 based on the reviewer comment here and earlier (#2) to articulate the motivation for using linear alkanes

instead of branched/cyclic alkanes and how they will not affect the findings from this work. To review the exact additions to the manuscript, see response to comment #2.

10. Page 7, last line: misspelled alkane

This has been corrected.

11. Page 8, paragraph 2: Gentner et al. (2012) estimated that branching and cyclization, which likely dominate I/SVOCs decrease SOA yields by a factor of around 3 (based on compiled chamber data available at that time). The assumption of a linear alkane SOM grid could consequently have a significant impact on SOA produced in this work. The authors point to Gentner and Caravaggio to justify that "alkanes" are the substantial fraction, but this somewhat obscures the fact that these alkanes are not linear, which may be important.

We completely agree with the reviewer on this comment, which is similar to those raised earlier (comments #2 and #9). The choice of a linear alkane to model the SOA formation from IVOCs should not have an effect on the model predictions in this work since the linear alkane choice was constrained to results from chamber experiments performed on combustion emissions from gasoline, diesel, and biomass burning sources. We have updated the text in Sections 2.2.2 and 5 to make this clear. To review the exact changes, we refer the reviewer/editor to our response to comment #s 2 and 9.

12. Page 14, paragraph 1: I'm a bit confused, if measured OA:OC is 1.8-2.1, but are the authors using 1.6?

An OA:OC ratio of 1.6 was used for the OC measured at the CSN (more urban) sites and an OA:OC ratio of 2.1 was used for the OC measured at the IMPROVE (more rural/remote) sites. The OA:OC ratio measured by Docherty et al. (2011) was 1.77, which was close to the value we used for the OC measured at the CSN site.

13. Figure 1 and discussion therefore: The terminology of splitting "IVOC" and "long alkane and aromatics" is a bit confusing, since the IVOCs are being modeled as long alkanes, and is comprised of alkanes and aromatics. I would recommended something more like "speciated" and "unspeciated", or "lumped". It's not totally clear to me what is a long alkane, and what is an IVOC, but perhaps I missed it in the methods?

IVOCs, as defined and described in Section 2.2.2, are unspeciated organic compounds found in combustion emissions with C* values mostly smaller than $10^6$ µg m$^{-3}$. In this work, the SOA formation from these IVOCs, regardless of their actual chemical makeup, was modeled by using a linear alkane as a surrogate, based on the work of Jathar et al. (2014). It is a separate matter that recent work with mobile source emissions has suggested that the IVOCs may in fact be a combination of branched/cyclic alkanes and aromatic compounds (e.g., for gasoline vehicle emissions suggested by Zhao et al. (2016)). Long alkanes, on the other hand, are emissions of alkanes that are currently in the emissions inventory but rarely include emissions of alkanes with C* values smaller than $10^6$ µg m$^{-3}$ (Pye and Pouliot, 2012). Given that the IVOCs and long alkanes are approximately separated in C* space (above and below $10^6$ µg m$^{-3}$) and treated separated in the emissions inventory, the IVOC label used in Figure 1 and throughout this work does not conflict with that of long alkanes. We agree with the reviewer that this needs to be made

clear in the manuscript. We have added some clarification in Section 2.2.1: "*Long alkanes as a precursor class includes linear, branched, and cyclic alkanes roughly up to a carbon number of $C_{13}$ and represent speciated alkanes present in existing emissions inventories. These long alkanes are distinct from the alkanes that might be present in SVOC and IVOCs. High-yield and lower-yield aromatics include all speciated aromatic compounds present in existing emissions inventories and, similar to the long alkanes precursor class, are distinct from the aromatics that might be present in SVOC and IVOCs.*". We have also added the following text in Section 2.2.2: "*Here, the IVOCs, as unspeciated organic compounds, are new SOA precursors added to the emissions inventory and regardless of their chemical makeup are distinct from the speciated precursors such as long alkanes and aromatics already present in existing emissions inventories.*".

14. Page 17, end of paragraph 1: What might explain this underestimation in SOA? Particularly given that the use of linear alkanes as proxies likely overestimates the SOA yield of some of these groups? Do the later changes to the model fix this regional underestimation?

The final version of the model used in this work (Base) still underestimates the SOA mass concentrations in southern California. The use of linear alkanes does not overestimate the SOA mass yield from IVOCs since the choice of the linear alkanes was constrained to chamber experiments (see earlier response to comments #2). As discussed in Section 5, the most likely reason for the underestimation might be a missing urban source or chemical pathway of SOA. It is also possible that the underestimation might be from the use of lower vapor loss rates in chambers than those recently suggested by Huang et al. (2018). We have added the following text in Section 5 to highlight the implications of this recent work: "*Recent work suggests that the vapor wall loss rates to the Teflon wall might be two or more times larger than the rates used in this work to develop the SOM parameters (Huang et al., 2018; Krechmer et al., 2016). The use of these faster rates will tend to increase the model predicted SOA mass concentrations and help explain the underpredictions with ambient measurements.*".

15. Page 17, end of paragraph 1: missing a period

This has been corrected.

16. Page 20, paragraph 3: "Our simulations imply that IVOCs as a bulk class of SOA precursors may not contribute substantially to ambient SOA levels." Again, I'm not quite sure what to make of this statement, as long alkanes may include species that would be considered IVOCs, so I'm a bit confused by imprecision in language around these compound classes.

The distinction between long alkanes, aromatics, and IVOCs has been made clear in the revised manuscript. See responses to reviewer comment #2 and #13. We have revised the statement the reviewer is referring to as follows: "*Our simulations imply that IVOCs might be as influential as SVOCs as a bulk class of SOA precursors but they were still less important than the traditional SOA precursors (that included long alkanes and aromatics) in contributing to ambient SOA levels.*".

17. Page 22: The authors discuss the impact of faster reaction times, but what about the impact of high volatility preventing wall loss. Does a C6 compound really suffer substantial wall loss,

given its ability to re-partition to the gas phase? I would expect most losses to be centered in the IVOC range (less reversibly absorbing to the walls).

It is important to remember that vapor wall losses affect both the precursor and the intermediate oxidation products that are still in the gas/vapor phase. Volatile precursors (such as the $C_6$ compound that the reviewer mentions) are not irreversibly lost to the walls but the oxidation products from the first few generations of chemistry might be in the gas/vapor phase as they are still too volatile to partition to the particle phase (with possibly in the IVOC range) and hence susceptible to loss to the chamber walls. In contrast, intermediate-volatility precursors have shorter chemical lifetimes and go through fewer steps of oxidation to form SOA, which makes them less susceptible to vapor wall losses.

18. Page 26, paragraph 2: The discussion of equations 1-4 is a bit unclear. It would be helpful to remind the reader of the implications of each equation.

We have added the following sentence to remind the reader of the functional form of the equation: "*To remind the reader, equations (1) and (2) assume a linear and logarithmic dependence respectively between the SOA mass concentration and the VOC:$NO_X$ ratio. Equations (3) and (4) assume a linear and logarithmic dependence respectively between the SOA mass concentration and the $NO_X$ concentration.*".

19. Figure 8, Table 4 and discussion thereof: What does it mean from a practical sense that the correlations here are so poor. While I acknowledge the biases and averages are not unreasonable, the model appears unable to capture the temporal variability of these measurements - is that simply due to poor resolution in emissions databases, or is there additional important complexity being ignored? The concern of course is that it might be telling us something more fundamental about the assumptions or applications of the model.

This is an important discussion point and we thank the reviewer for the comment. To expand on what the reviewer is suggesting, the poor correlations are indicative of poor model skill where the model is unable to accurately capture the temporal or spatial variability in the measurements. However, the model skill showcased here, particularly for OA, is not very different than that produced by other models in the literature. For example, Baker et al. (2015) in an application of the Community Multiscale Air Quality (CMAQ) model reported an $R^2$ of 0.0036 for their base model and an $R^2$ of 0.10 for a sensitivity simulation for OA over all CSN and IMPROVE sites in southern California. In this work, we report $R^2$ values of 0.13 and 0.079 over the CSN and IMPROVE networks respectively, which are slightly better than those reported by Baker et al. (2015) in the same geographical area but a different time period. However, Murphy et al. (2017) in an application of a similar CMAQ model over the contiguous United States reported an higher $R^2$ of 0.26 for OA over all CSN and IMPROVE sites. Similarly, Ahmadov et al. (2012) in an application of the Weather Research and Forecasting - Chemistry (WRF-Chem) model reported an $R^2$ between 0.30 and 0.37 for OA over CSN and IMPROVE sites in the eastern United States. If representative, the Murphy and Ahmadov results suggest that the emissions and chemistry of OA (and perhaps even the meteorology) are not well represented by models in the southern California region. There are numerous reasons for why the model skill may be so poor in this region. One of the reasons, as the reviewer suggests, could be the poor spatial and temporal resolution offered in the emissions inventory. But we have not explored this aspect in our work. The reviewer's comment about model skill is discussed in Section 4.2: "*The model skill,*

*captured by the $R^2$ values, for all model simulations at both the CSN and IMPROVE sites was quite poor, but still slightly better than that found in earlier work for the southern California region with the CMAQ model (Baker et al., 2015). However, the model skill was much worse than that reported in earlier work with CMAQ (e.g., Murphy et al. (2017)) and WRF-Chem (e.g., Ahmadov et al. (2012)) over regions other than southern California, suggesting that there might be missing emissions sources and/or chemical pathways or meteorological considerations that contribute to the poor model skill in southern California.".*

20. Figure 9: It's not clear why an O:C of 0.078 was chosen. I imagine it comes from the measurements, but it's not obvious where this is stated. Update: I see it is stated later, this should be brought forward to discussion of the figure or the caption.

We have added the necessary citation to the caption of Figure 9: "The three different predictions show results from the Base simulations for OA assuming no change, the POA O:C was fixed to 0.078 *based on the measurements of Docherty et al. (2011)*, and no POA.". We also added detail to Section 4.4 to link the various model predictions discussed in the text to those in Figure 9. For example, "For the Base simulations *(shown as orange box plots)*, model predictions of H:C were significantly overpredicted and those for O:C were significantly underpredicted although the predictions did capture dips in the H:C and the peaks in the O:C ratios in the mid-afternoon, coincident with peak photochemical activity.". Similar changes were made elsewhere in Section 4.4. when referencing the predictions in Figure 9.

21. Page 34 paragraph 3: Given how big a role vapor wall loss correction plays in the model, it would be helpful to have some discussion of how exactly it is being corrected for. This manuscript just references previous studies, but it warrants some overview here.

We rely on the SOM parameters developed in Cappa et al. (2016) based on the methods described in Zhang et al. (2014), to account for the influence of vapor wall losses in Teflon chambers. As these methods have been previously described in the earlier work, we do not feel the need to include a detailed description of how the wall losses were modeled, at the expense of increasing the length of this manuscript. We have added the following text in Section 2.2.1 when describing the wall loss-corrected SOM parameters for the first time: "*
[revised manuscript text omitted]

| Page 25: [1] Formatted | Shantanu Jathar | 2/1/19 4:05:00 PM |
|---|---|---|

Subscript

---

## Author Response (AR2)

We thank the reviewer for reading through our response, changes to the manuscript, and the additional comments, despite the length of the manuscript and the response to reviewer comments. We also thank the editor for being very responsive to our submissions. Like the earlier response, reviewer comments are in regular black, our response is in blue, text from the manuscript is in red, and additions/updates are in *italic magenta*. We are happy to include any more suggestions that the reviewer and editor have.

(1) It is still hard to extract the "big picture" message from the paper, though this is mitigated to some extent by the sheer magnitude of the effort presented in the manuscript. In the response to reviews, the authors note 5 main points from this paper. Ideally it would be more like 2-3.

If we had to choose 2 main points that are unique to our work, we would choose the primary take-aways to be: (i) S/IVOCs, after accounting for the influence of vapor wall losses, do not contribute as much to the SOA burden as traditional VOC precursors (e.g., aromatics), (ii) accounting for the influence of $NO_X$ may increase SOA mass concentrations in high $NO_X$/urban regions, and (iii) updates included in this work seem to improve the model-measurement comparison for OA mass and composition in southern California. We have edited the abstract slightly (removed sentences) in the hope to sharpen its focus on the three points mentioned above. See tracked changes in the revised submission.

(2) The entirety of section 4 on model evaluation could be moved to the supporting information. This might make the big picture results stand out more, but keeping this section in the main text should not hamper publication.

The broad model evaluation undertaken in this work that included comparisons for SOA precursors and OA mass and composition is required to understanding the accuracy and limitations in modeling OA in chemical transport models. Hence, we have decided to keep this section within the main text, despite the length of the manuscript.

(3) I think Figure 5 shows an important result that is maybe not stressed enough. The simulations in this paper push pretty hard on POA partitioning and SOA formation. Even with all of that pushing, the total OA only varies by a factor of ~2 at both high and low NOx. This suggests that while there are uncertainties to work out (e.g., I/SVOC emissions and volatility), OA is constrained reasonably well.

We thank the reviewer for this insightful comment. We have added the following text to the discussion - "*The emissions inputs and chemical treatment for OA was varied substantially in the sensitivity simulations performed in this work. Yet, the simulations seemed to change the OA by less than a factor of 2 suggesting that the model framework, except for the treatment of $NO_X$, was generally reasonable in constraining in the total OA mass concentrations in southern California.*" and the following text to the abstract "*The updated model's performance against measurements combined with the results from the sensitivity simulations suggest that the OA mass concentrations in southern California are constrained within a factor of two.*".

(4) Page 10 describes the IVOC:NMOG ratios applied in the model. The specific ratios used in

the model differ from Zhao et al's measurements. My understanding is that this is done to tune SOA mass formation - the IVOC:NMOG ratio needs to be adjusted to account for the use of a single SOA surrogate for each SOM grid. This tuning likely has an impact on model transferability in both space and time. If the combustion IVOC emissions mix is either spatially or temporally variable, the model may be biased when applied outside of the LA-Riverside domain or for predicting future cases.

Yes, the reviewer is correct in that the IVOC:NMOG ratio and the surrogate used to model the SOA formation are coupled, i.e., if one were changed, the other would change too to be consistent with the chamber data. The two paragraphs on page 10 try to explain the differences in the IVOC:NMOG ratios and the SOA parameterizations for IVOCs used in this work to the way they were determined and modeled in Zhao et al. (2015, 2016). Overall, the treatment in this work and the work of Zhao et al. (2015, 2016) is consistent if we consider median model-measurement comparison for SOA at the end of the chamber experiment. Both treatments are likely to produce differences in the time-dependent evolution of SOA as well as in the relative contributions of the different precursors to the total SOA. We have made a quick note about that on page 10: "*In a future version of the model, we will aim to include the IVOC emissions estimates of Zhao et al. (2015, 2016) and update the SOA parameterizations accordingly. It is likely that these might slightly alter the spatiotemporal distribution of IVOC SOA in the modeled domain.*".

(5) I still disagree that volatile chemical products are a major source of IVOCs that could produce enough SOA to reach mass closure with measurements. I think this is treated appropriately in the text but is stated too strongly in the abstract.

We have removed the volatile chemical product-related hypothesis in the abstract.

[revised manuscript text omitted]